# Structure and immunogenicity of a stabilized HIV-1 envelope trimer based on a group-M consensus sequence

Kwinten Sliepen [1,15], Byung Woo Han [2,3,16], Ilja Bontjer[1,15], Petra Mooij[4], Fernando Garces[2,12], Anna-Janina Behrens[5,13], Kimmo Rantalainen[2], Sonu Kumar [2], Anita Sarkar [2], Philip J.M. Brouwer[1], Yuanzi Hua[2], Monica Tolazzi[6], Edith Schermer[1], Jonathan L. Torres [2], Gabriel Ozorowski [2], Patricia van der Woude[1], Alba Torrents de la Peña[1], Mariëlle J. van Breemen[1], Juan Miguel Camacho-Sánchez [1], Judith A. Burger[1], Max Medina-Ramírez[1], Nuria González [7], Jose Alcami[7], Celia LaBranche[8], Gabriella Scarlatti[6], Marit J. van Gils [1], Max Crispin[5,14], David C. Montefiori[8], Andrew B. Ward [2], Gerrit Koopman[4], John P. Moore [9], Robin J. Shattock[10], Willy M. Bogers[4], Ian A. Wilson [2,11,16] & Rogier W. Sanders[1,9,16]

Stabilized HIV-1 envelope glycoproteins (Env) that resemble the native Env are utilized in vaccination strategies aimed at inducing broadly neutralizing antibodies (bNAbs). To limit the exposure of rare isolate-specific antigenic residues/determinants we generated a SOSIP trimer based on a consensus sequence of all HIV-1 group M isolates (ConM). The ConM trimer displays the epitopes of most known bNAbs and several germline bNAb precursors. The crystal structure of the ConM trimer at 3.9 Å resolution resembles that of the native Env trimer and its antigenic surface displays few rare residues. The ConM trimer elicits strong NAb responses against the autologous virus in rabbits and macaques that are significantly enhanced when it is presented on ferritin nanoparticles. The dominant NAb specificity is directed against an epitope at or close to the trimer apex. Immunogens based on consensus sequences might have utility in engineering vaccines against HIV-1 and other viruses.

[1] Department of Medical Microbiology, Amsterdam Infection & Immunity Institute, Amsterdam UMC, University of Amsterdam, Meibergdreef 9, Amsterdam 1105AZ, The Netherlands. [2] Department of Integrative Structural and Computational Biology, Scripps CHAVI-ID, IAVI Neutralizing Antibody Center and Collaboration for AIDS Vaccine Discovery (CAVD), The Scripps Research Institute, La Jolla, CA 92037, USA. [3] Research Institute of Pharmaceutical Sciences, College of Pharmacy, Seoul National University, Seoul 08826, Korea. [4] Department of Virology, Biomedical Primate Research Centre, 2280 GH Rijswijk, The Netherlands. [5] Oxford Glycobiology Institute, Department of Biochemistry, University of Oxford, Oxford OX1 3QU, UK. [6] Viral Evolution and Transmission Unit, Division of Immunology, Transplantation and Infectious Diseases, IRCCS San Raffaele Scientific Institute, Milan 20132, Italy. [7] AIDS Immunopathology Unit, Instituto de Salud Carlos III, Madrid 28220, Spain. [8] Department of Surgery, Duke University Medical Center, Durham, NC 27710, USA. [9] Department of Microbiology and Immunology, Weill Medical College of Cornell University, New York, NY 10021, USA. [10] Section of Virology, Division of Infectious Diseases, Department of Medicine, Imperial College London, Norfolk Place, London W2 1PG, UK. [11] The Skaggs Institute for Chemical Biology, The Scripps Research Institute, La Jolla, CA 92037, USA. [12] Present address: Department of Therapeutics Discovery, Amgen Research, Amgen Inc., 1 Amgen Center Drive, Thousand Oaks, CA 91320, USA. [13] Present address: New England Biolabs Inc., 240 County Road, Ipswich, MA 01938, USA. [14] Present address: Centre for Biological Sciences and Institute for Life Sciences, University of Southampton, Southampton SO17 1BJ, UK. [15] These authors contributed equally: Kwinten Sliepen, Ilja Bontjer. [16] These authors jointly supervised this work: Ian A. Wilson, Rogier W. Sanders, Byung Woo Han. Correspondence and requests for materials should be addressed to B.W.H. (email: bwhan@snu.ac.kr) or to I.A.W. (email: wilson@scripps.edu) or to R.W.S. (email: r.w.sanders@amc.uva.nl)

The enormous diversity of human immunodeficiency virus (HIV)-1 strains is a major obstacle for the development of a broadly protective vaccine[1,2]. Diversity is highest in the envelope glycoprotein complex (Env), the target for neutralizing antibodies (NAbs), where amino acid sequences can differ by up to 35% between subtypes. For comparison, a change of 2% at the amino acid level in influenza hemagglutinin can render vaccine-induced immune responses ineffective, requiring yearly updates of the influenza vaccine[1]. Thus HIV-1 diversity makes it extremely difficult to induce broadly reactive NAbs (bNAbs) by an Env immunogen from any one particular HIV-1 isolate.

Moreover, even NAb responses against the sequence-matched neutralization-resistant (Tier 2) virus are usually not induced by gp120 or non-native Env trimers and were facilitated by the design of stable soluble mimics of the native Env trimer, such as BG505 SOSIP.664[3–5]. This prototype native-like trimer contains an extra intermolecular disulfide bond (SOS) to link gp120 and gp41 and a point substitution (I559P, i.e., IP) to aid assembly of the gp41 subunits into their pre-fusion form[4,6,7]. High-resolution X-ray and cryo-electron microscopic structures of BG505 SOSIP.664[8] have enabled further improvement and stabilization of BG505 SOSIP trimers and facilitated the generation of SOSIP trimers from other HIV-1 strains and subtypes[9]. However, such trimers might not be expected to represent stand-alone vaccines, since these immunogens predominantly induced isolate-specific NAbs and sporadic and/or weak heterologous Tier 2 NAb responses at best, even when used in combination or sequential immunization regimens[10,11].

Considering HIV-1's diversity, native-like trimers of any specific viral isolate might not be the optimal platform for inducing bNAbs. Immunogens based on consensus sequences might be more appropriate, since a consensus sequence is usually closer to circulating isolates than circulating isolates are to one another[2]. Furthermore, rare isolate-specific antigenic residues/determinants are "averaged away" in the consensus sequence, which might favor more cross-reactive responses over isolate-specific responses[2]. However, these semi-artificial consensus sequences might not fold into a native-like Env structure efficiently, because important isolate-specific interactions within the trimer might be removed in generating a consensus sequence. Recombinant non-native Env protein immunogens based on consensus sequences have previously been generated[2,12–15]. However, such non-native immunogens are unlikely to induce NAbs against quaternary structure-dependent epitopes that are only present on the native trimer including, but not limited to, ones that target the trimer apex.

Here we describe the generation, crystal structure, and immunogenic properties of a native-like Env (SOSIP) trimer based on a group M consensus sequence. The ConM SOSIP trimer induces NAbs against the autologous Tier 1A virus and related Tier 1B ConS virus. In addition, these responses target the trimer apex and are enhanced when the trimers are presented on nanoparticles. This study shows that native-like HIV-1 Env trimers can be generated from consensus sequences and such immunogens might be suitable vaccine components to prime and/or boost desirable NAb responses.

## Results

### A group M consensus Env sequence yields native-like trimers.
We selected the last published group M consensus sequence (from 2004; www.hiv.lanl.gov; alignment ID: 104CP2; Fig. 1a; Supplementary Fig. 1), which is a consensus of the consensus sequences of each clade in group M, from hereon termed ConM. We first generated a construct based on the SOSIP.v4.2 design (Fig. 1b)[16], but to improve the quality, stability, and yield of

ConM SOSIP.v4.2, we added a second intermolecular disulfide bond between residues 73 in gp120 and 561 in gp41 (SOSIP.v5.2)[5] and a number of amino acid changes in the trimer interface derived from the BG505 sequence, collectively termed TD8[17]. We named the construct that incorporated all of these changes ConM SOSIP.v7 (Fig. 1b, Supplementary Table 1). The interprotomer disulfide bond included in the SOSIP.v6 design was not incorporated because of its adverse effects on trimer yields[5].

Initial screening of unpurified ConM SOSIP.v4.2, SOSIP.v5.2, and SOSIP.v7 proteins from transiently transfected 293T cell supernatants showed that all constructs produced trimers, but the SOSIP.v7 construct yielded higher-quality trimers as demonstrated by more efficient binding of quaternary-dependent bNAbs PG16, PGT145, PGT151, and 35O22 in an enzyme-linked immunosorbent assay (ELISA; Supplementary Fig. 2a, b). The ConM SOSIP.v7 construct also resulted in higher trimer yields (~4.9 mg/L) than SOSIP.v4.2 (~0.9 mg/L) and SOSIP.v5.2 (~0.9 mg/L) when purified by affinity chromatography using trimer-specific PGT145 bNAb (Table 1). Similarly, when the ConM proteins were purified from transiently transfected 293S cells by 2G12-affinity chromatography followed by size exclusion chromatography (SEC), the SOSIP.v7 construct resulted in higher trimer yields (77% trimers, ~1.0 mg/L) compared to SOSIP.v4.2 (35%, ~0.2 mg/L) and SOSIP.v5.2 (65%, ~0.7 mg/L) (Fig. 1c).

All purified ConM SOSIP trimer versions were completely cleaved (Supplementary Fig. 2c), but ConM SOSIP.v7 interacted more efficiently with quaternary structure-dependent bNAbs PG16, PGT145, PGT151, and 35O22 and showed decreased binding to undesirable V3-directed non-NAb 19b (Supplementary Fig. 2d). Negative-stain electron microscopy (NS-EM) revealed that ~100% of the ConM SOSIP.v7 trimers had a native-like morphology (Fig. 1d, Table 1). Differential scanning calorimetry (DSC) analyses showed that ConM SOSIP.v7 displayed high thermostability that was comparable to BG505 SOSIP.664 (midpoint of thermal denaturation ($T_m$) = 67.8 °C and 68.1 °C, respectively) (Fig. 1e, Table 1)[4]. Moreover, the majority of the glycans on ConM SOSIP.v7 trimers are oligomannose (68%), in particular $Man_9GlcNAc_2$ (25%) and $Man_8GlcNAc_2$ (12%) (Fig. 1f, Table 1), which very closely matches other soluble native-like Env trimers[18–22], and is consistent with the strong binding of glycan-dependent bNAbs (see above and below).

By all the evaluated criteria, we conclude that the ConM SOSIP.v7 construct produced high-quality, native-like trimers that can be purified at high yield in different expression systems, across different laboratories, and using different purification methods.

### ConM SOSIP.v7 engages germline precursors of V2-apex bNAbs.
Env immunogens that interact with bNAbs do not necessarily bind the germline precursors of these bNAbs, in which case they are unlikely to induce similar bNAbs by vaccination[23–27]. We tested the ability of ConM SOSIP.v7 trimers to bind to the inferred germline precursors of several bNAbs (gl-bNAbs) in a Ni-NTA His-capture ELISA, focusing on V2-apex bNAbs. As comparators, we used the BG505 SOSIP.v5.2 trimer and the BG505 GT1.1 trimer, an updated germline-targeting immunogen derived from the BG505 GT1 trimer that was specifically engineered to engage VRC01-class gl-bNAbs and enhance binding to V2-apex gl-bNAbs[28,29].

The BG505 SOSIP.v5.2 trimer interacted well with gl-PG9 and gl-CH01 and also weakly with gl-PG16, consistent with previous observations using BG505 SOSIP.664 (Fig. 1g; ref. [24]); however, the GT1.1 trimer bound more strongly to gl-PG9, gl-PG16, and gl-CH01 than its BG505 SOSIP precursor, as reported previously (Fig. 1g; ref. [29]). The ConM SOSIP.v7 trimer engaged gl-PG9 and

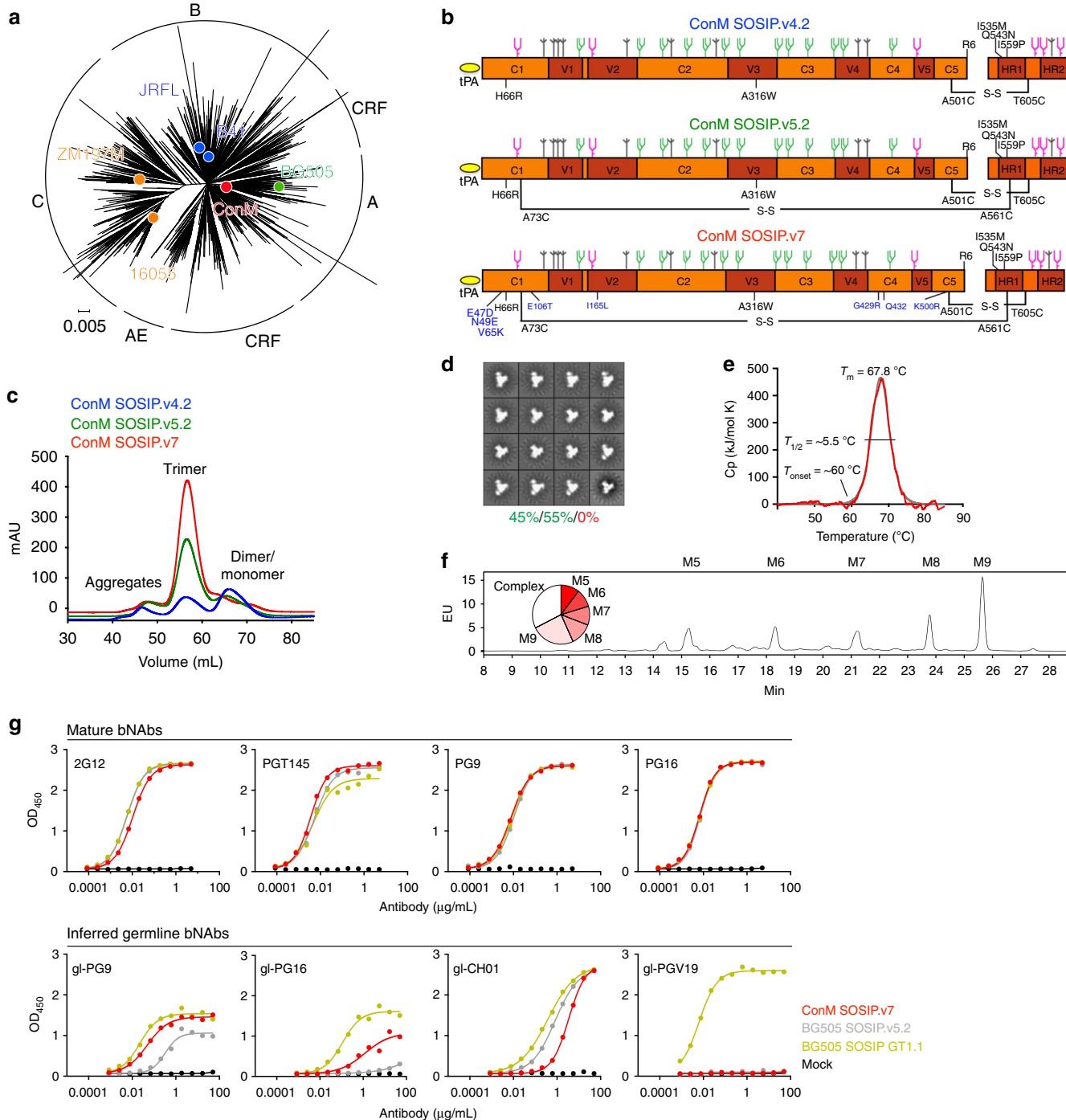

gl-PG16 with higher affinity compared to the BG505 SOSIP.v5.2 trimer but not as strongly as the GT1.1 trimer. Furthermore, the ConM SOSIP.v7 trimer was able to engage gl-CH01, albeit with lower affinity compared to the other two trimers. Only the engineered GT1.1 trimer interacted with the VRC01-class bNAb gl-PGV19 (Fig. 1g). We conclude that, even without specific germline-targeting modifications, the ConM SOSIP.v7 trimer was able to engage the inferred germline precursors of several V2-apex bNAbs.

**ConM SOSIP.v7 is structurally similar to other Env trimers.** Native-like trimers from HIV-1 clades A, B, C, and G have been characterized extensively by biochemical and structural

approaches[30–36], and recently, an Env trimer based on the consensus sequence of several clade C Env sequences was described[37]. A high-resolution structure would inform future structure-based design efforts that use ConM SOSIP.v7 as the platform. Therefore, we determined the crystal structure of ConM SOSIP.v7 in complex with bNAbs PGT124 and 35O22 at 3.9 Å resolution by molecular replacement (MR) method using the crystal structure of BG505 SOSIP.664 in complex with Fabs 3H+109L and 35O22 (PDB ID: 5CEZ) as a MR phasing model[32] (Fig. 2a, Table 2).

Overall, the ConM SOSIP.v7 trimer has high structural similarity with trimers from clade A (BG505 SOSIP.664), clade C (16055 NFL TD CC), and clade G (X1193.c1 SOSIP.664), as well as membrane-derived native trimers from clade B (JRFL Env

**Fig. 1** Design, screening, and biophysical characterization of ConM SOSIP trimers. **a** Maximum likelihood tree generated with all Env sequences of the M group from the Los Alamos HIV database webalignment 2011 ($n = 3654$) and the parental Env sequences of five previously described SOSIP trimers[4, 17, 72, 81] and ConM. Different subtypes and genetic distance bar (substitutions per sequence position) are indicated. **b** Linear representation of the three ConM SOSIP variants. Top: design of SOSIP.v4.2 with the SOS-bond (A501C-T605C), improved furin cleavage site (R6), IP (I559P), trimer-stabilizing mutations (H66R and A316W), and improved trimerization mutations (I535M and Q543N)[4, 16]. Middle: design of SOSIP.v5.2 with an extra intermolecular disulfide bond (A73C-A561C)[5]. Bottom: design of SOSIP.v7 with TD8 mutations in blue[17]. The assignment of glycans was based on the analysis of BG505 SOSIP.664 reported in ref. [41] but may be different for ConM SOSIP trimers. **c** Size exclusion chromatographic profiles of 2G12-purified ConM SOSIP trimers (untagged) expressed in 293S cells on a Superdex200 16/60 column. **d** 2D class average of negative-stain electron microscopic analyses of PGT145-purified ConM SOSIP.v7 trimer (D7324-tagged) expressed in 293F cells. Percentages of closed and open native-like are depicted in green and non-native in red. **e** Thermostability derived from differential scanning calorimetry (DSC) of PGT145-purified ConM SOSIP.v7 trimer (D7324-tagged) expressed in 293F cells. The DSC parameters midpoint of thermal transition ($T_m$), starting temperature of unfolding ($T_{onset}$), and width of the transition at half peak height ($T_{1/2}$) are indicated. **f** Hydrophilic interaction liquid chromatography-ultra performance liquid chromatographic spectra of N-linked glycans derived from PGT145-purified ConM SOSIP.v7 trimer (D7324-tagged) in 293F cells. Peaks corresponding to oligomannose glycans ($Man_{5-9}GlcNac_2$) are labeled M5–M9. The smaller peaks that are not highlighted correspond to complex or hybrid-type glycans. The relative abundances of M5–M9 and complex glycans are summarized in a circle diagram. **g** Ni-NTA-capture enzyme-linked immunosorbent assay with PGT145-purified ConM SOSIP.v7 trimer and the comparator trimers BG505 SOSIP.v5.2 and GT1.1 (all His-tagged) expressed in 293F cells against a panel of (gl-)bNAbs

**Table 1 Biophysical characterization of stabilized ConM SOSIP trimers**

| SOSIP version[a] | | | v4.2 | v5.2 | v7 | v7-ferritin |
|---|---|---|---|---|---|---|
| Production | | Yield (mg/L)[b] | 0.9[c] | 0.9[c] | 4.9[c] | 3.5 |
| | | Yield (mg/L)[d] | 0.2[e] | 0.7[e] | 1.0[e] | ND |
| Morphology | NS-EM[b] | Native-like trimers (%) | ND | ND | ~100[c] | ND |
| | | Closed native-like trimers (%) | ND | ND | ~45[c] | ND |
| | DLS[b] | $R_h$ (nm) | ND | ND | 6.4[c] | 13.4 |
| | | Pd (%) | ND | ND | 12.5[c] | 59.9 |
| Thermostability[f] | DSC[b] | $T_m$ (°C) | 65.1[c] | ND | 67.8[c] | 72.0 |
| | DSC[d] | $T_m$ (°C) | 65.6[e] | 68.8[e] | 66.7[e] | ND |
| Glycan composition | HILIC-UPLC[b] | $Man_8$ (%) | 21[c] | ND | 12[c] | 20 |
| | | $Man_9$ (%) | 23[c] | ND | 25[c] | 24 |
| | | Oligomannose (%) | 75[c] | ND | 68[c] | 75 |

*DLS* dynamic light scattering, *DSC* differential scanning calorimetry, *HILIC-UPLC* hydrophilic interaction liquid chromatography-ultra performance liquid chromatography, *ND* not determined, *NS-EM* negative-stain electron microscopy
[a]A linear representation and an overview of the modifications made to the trimer variants is shown in Figs. 1b and 2a and Supplementary Table 1
[b]Data were derived from 293F cell-expressed and PGT145-purified SOSIP trimers
[c]Data were derived from SOSIP trimers containing a D7324-tag
[d]Data were derived from 293S cell-expressed and 2G12/SEC-purified SOSIP trimers
[e]Data were derived from SOSIP trimers without a tag
[f]$T_m$ values that were obtained with D7324-tagged SOSIP trimers were up to 0.3 °C higher than those without a tag

from which the cytoplasmic tail was removed (ΔCT)), with $C_\alpha$ root mean square deviations of 0.80 Å, 0.60 Å, 0.79 Å, and 1.05 Å, respectively (Fig. 2b)[30–32,34–36]. The overall structural similarity confirms that the ConM SOSIP.v7 trimer, based on a consensus sequence, mimics the native Env trimer conformation. Interactions between ConM SOSIP.v7 and bNAbs PGT124 and 35O22 are similar to those reported for other trimers in that PGT124 recognizes the N332 glycan and the $^{324}$GDIR$^{327}$ motif at the V3 loop base and 35O22 interacts at the gp120/gp41 interface (Supplementary Fig. 3a, b)[34–36,38–40]. All of the TD8 residues (E47D, N49E, V65K, E106T, I165L, G429R, Q432, and K500R) could be resolved in our crystal structure and their side chains were similarly oriented as in the clade A BG505 SOSIP.664 structure (PDB ID: 5CEZ, Supplementary Fig. 4a). The intersubunit C73-C561 disulfide bond could not be confirmed in the structure because the region around residue C561 lacked interpretable electron density. On each ConM SOSIP.v7 monomer, 49 glycan saccharide residues of glycans at 16 of the 30 predicted that Asn-X-Ser/Thr sequons had interpretable electron density (Supplementary Table 2 and Supplementary Fig. 4b)[41].

**ConM SOSIP.v7 displays few rare antigenic determinants**. To further study the antigenic surface of the ConM SOSIP.v7 trimer, we analyzed the conservation of each residue among all the HIV-1 envelope glycoprotein sequences ($n = 6112$) available from the Los Alamos HIV database (http://www.hiv.lanl.gov/), as present in 2017. When we compared amino acid conservation of each equipositional residue of ConM and BG505, 65 residues exhibit differences in the amino acid conservation. ConM contains 14 residues that are rarer than equipositional residues of BG505 while BG505 contains 51 rarer residues. Considering the trimeric assembly of Env proteins, the ConM trimer would contain many fewer rarer residues than the BG505 trimer (Fig. 2c and Supplementary Table 3). Furthermore, analysis of the ConM *env* sequence using the Glycan Shield Mapping tool[42] showed that ConM does not lack potential N-linked glycosylation sites that might cause holes in the glycan shield, in contrast to the protypic BG505 *env* sequence (Fig. 2d).

Together, these data imply that the ConM trimer displays considerably less rare antigenic residues/determinants compared with the BG505 trimer, a property that might facilitate the induction of cross-reactive NAbs.

**ConM SOSIP.v7 nanoparticles activate cognate B cells**. Particulate display of antigens can improve immunogenicity by enhancing B cell receptor (BCR) cross-linking and B cell activation[43]. Indeed, the immunogenicity of BG505 SOSIP.664 trimers can be enhanced by presenting them on ferritin nanoparticles displaying eight trimers per particle[44]. Therefore, we genetically

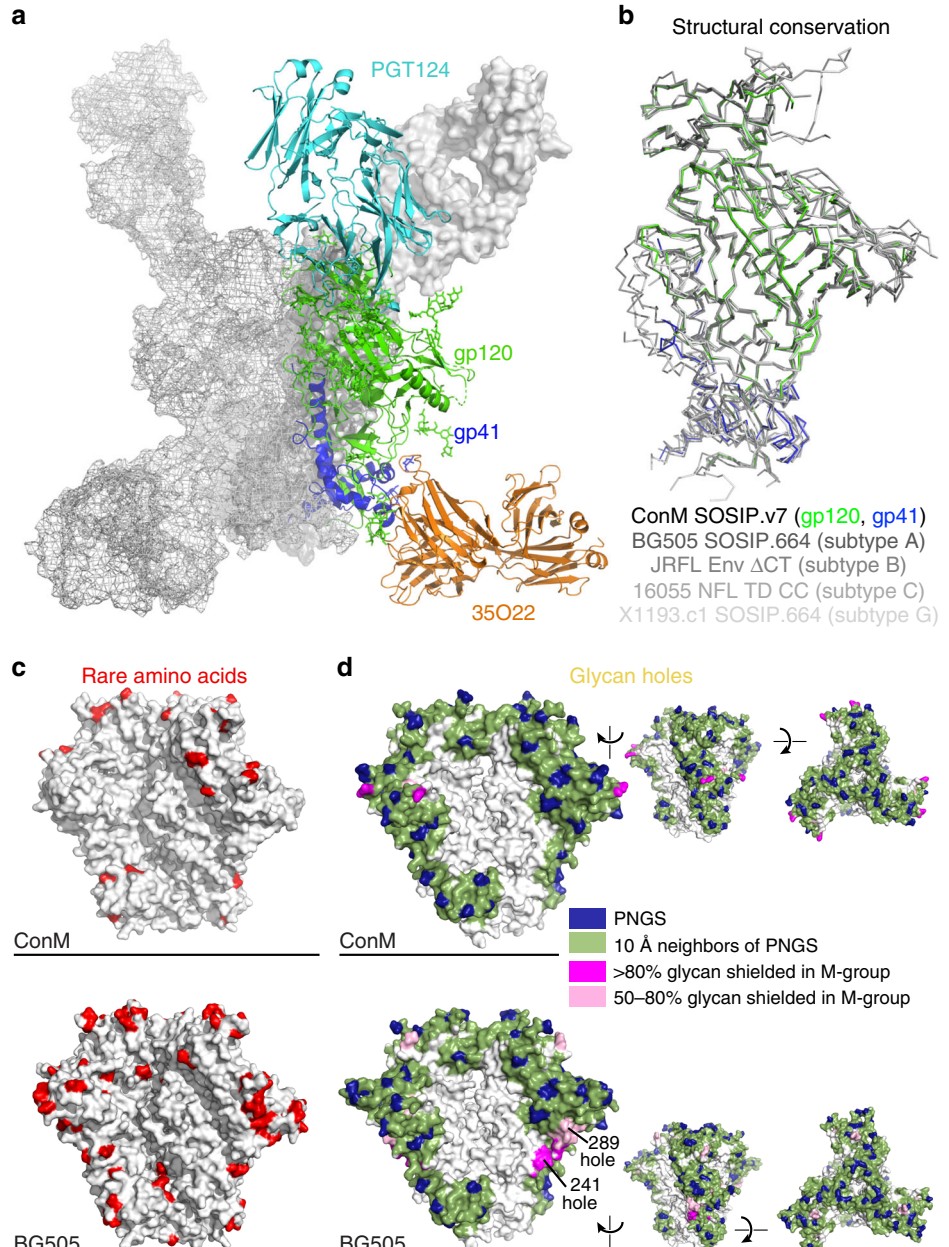

**Fig. 2** The consensus sequence-based ConM SOSIP.v7 trimer shares structural similarity to other Env structures. **a** Crystal structure of ConM SOSIP.v7 trimer (gp120 green, gp41 blue) in complex with Fabs PGT124 (cyan) and 35O22 (orange) at 3.9 Å resolution. For clarity, two other ConM SOSIP.v7 protomers are represented as white surface and mesh. **b** Superimposition of the soluble ConM SOSIP.v7 monomer (gp120 green, gp41 blue), clade A BG505 SOSIP.664 (PDB: 5CEZ), clade B JRFL Env ΔCT (PDB: 5FUU), clade C 16055 NFL TD CC (PDB: 5UM8), and clade G X1193.c1 SOSIP.664 (PDB: 5FYJ). **c** Comparison of rarer amino acid residues of ConM and BG505. After analyzing amino acid conservation for each residue among all the HIV-1 envelope glycoprotein sequences ($n = 6112$) available from the Los Alamos HIV database 2017, amino acid residues in ConM and BG505 that are rarer than equipositional residues in BG505 and ConM, respectively, are represented in red. There are 14 and 51 rarer residues in each ConM and BG505 protomer, respectively. **d** Visualization of the glycan shield of ConM Env, containing no glycan holes (top), and the prototypic BG505 Env, which contains the 241/289 glycan hole (bottom). Glycan shields were modeled using the Glycan Shield Mapping tool[42]

fused ConM SOSIP.v7 to *Helicobacter pylori* ferritin (Fig. 3a) and purified these nanoparticles by PGT145 purification from the supernatant of transiently transfected 293F cells (yield of ~3.5 mg/L) (Table 1). ELISA showed that the ferritin scaffold did not dramatically change the antigenicity of ConM SOSIP.v7 trimers, since most bNAbs (2G12, VRC01, PGT121, PGT135, and PG16) interacted with the ConM trimers on the nanoparticles (Supplementary Fig. 5a) and these data were corroborated by biolayer interferometry (BLI) experiments (Supplementary Fig. 6). However, a minority of the ConM SOSIP.v7-ferritin material was

uncleaved at the gp120/gp41 junction, despite the use of increased amounts of furin (see "Methods" ; Supplementary Fig. 5b). Similar observations were also made with ferritin nanoparticles displaying SOSIP trimers from other isolates[44,45]. Thus furin access to its substrate cleavage site appears to be hindered to some extent when Env trimers are presented on ferritin particles.

Negative stain-EM showed that ConM SOSIP.v7-ferritin formed particles with a diameter of 30–40 nm consisting of a spherical ferritin core from which eight trimer spikes protruded, although some incompletely formed particles were also visible in

**Table 2 Data collection and refinement statistics**

|  | ConM SOSIP.v7 in complex with PGT124 and 35O22 |
|---|---|
| *Data collection* |  |
| Space group | P6₃ |
| Cell dimensions |  |
| $a, b, c$ (Å) | 127.59, 127.59, 315.51 |
| $\alpha, \beta, \gamma$ (°) | 90, 90, 120 |
| Resolution (Å) | 50.0-3.90 (4.04-3.90)a |
| Observations | 412,746 |
| Unique reflections | 23.583 (1613) |
| $R_{sym}$ | 0.15 (1.07) |
| $I / \sigma I$ | 15.7 (1.0) |
| $CC_{1/2}$ | 0.95 (0.70) |
| Completeness (%) | 89.0 (61.7) |
| Redundancy | 17.5 (12.3) |
| *Refinement* |  |
| Resolution (Å) | 30.0-3.9 |
| No. of reflections | 22,323 |
| $R_{work}/R_{free}$ (%) | 24.7/29.6 |
| No. of atoms |  |
| gp120 | 3483 |
| gp41 | 1043 |
| PGT124 | 3.349 |
| 35O22 | 3346 |
| Glycans | 625 |
| *B-factors* |  |
| gp120 | 233.5 |
| gp41 | 222.7 |
| PGT124 | 261.8 |
| 35O22 | 283.0 |
| Glycans | 272.6 |
| R.m.s. deviations |  |
| Bond lengths (Å) | 0.007 |
| Bond angles (°) | 1.312 |

aValues in parentheses are for the highest-resolution shell

the raw micrographs (Fig. 3b and Supplementary Fig. 5c). The glycan composition of the ferritin-presented ConM SOSIP.v7 trimers was similar to that of the soluble trimers (Fig. 3c, Table 1) and the dynamic light scattering (DLS) analyses confirmed the dimensions that were observed in NS-EM with a hydrodynamic radius ($R_h$) of 13.4 nm for ConM SOSIP.v7-ferritin and 6.4 nm for single ConM SOSIP.v7 trimers (Fig. 3d, Table 1). The polydispersity was relatively high (>15%)[46], consistent with the presence of incompletely formed particles. The ConM SOSIP.v7-ferritin particle displayed an increased thermostability compared to free ConM SOSIP.v7 trimers ($T_m = 72.0$ °C versus $T_m = 67.8$ °C, respectively), but the melting peak of the nanoparticles was slightly broader than that of the trimer only ($T_{1/2} = 6.3$ °C versus $T_{1/2} = 5.5$ °C), because of the concurrent melting of the ferritin nanoparticle and/or because of the heterogeneity among the trimers on the ferritin nanoparticles (Fig. 3e, Table 1).

We also addressed the thermostability of antigenic determinants. Accordingly, we incubated ConM trimers and ConM trimers on ferritin nanoparticles for an hour at 4 °C (negative control), 37 °C (mimicking vaccination), and at 60 °C, 68 °C, and 72 °C (mirroring the unfolding events in the DSC profile), prior to antigenicity analysis using BLI (Fig. 3f, Supplementary Fig. 6).

Single ConM SOSIP.v7 trimers interacted efficiently with the quaternary-specific bNAb PGT145, but the binding signal was reduced after incubation at 60 °C ($T_{onset}$), consistent with initiation of unfolding and the dissociation of trimer interactions, and was abrogated after incubation at 68 °C or 72 °C. The reverse result was obtained with 19b for which the binding steadily increased with temperature, consistent with exposure of its linear epitope upon protein unfolding. F105 binding was increased after incubation at 60 °C, consistent with trimer dissociation, but decreased after incubation at 68 °C or 72 °C, in line with loss of gp120 integrity.

The nanoparticle-displayed ConM SOSIP.v7 trimers showed enhanced binding to 2G12 and PGT145 bNAbs compared to the soluble trimers at 4 °C and 37 °C, probably due to increased avidity of the nanoparticle-displayed epitopes (Fig. 3f). At 60 °C ($T_{onset}$) and 68 °C ($T_m$ of ConM SOSIP.v7), PGT145 binding was (mostly) retained for ConM SOSIP.v7-ferritin, while it was strongly decreased for the soluble trimers (Fig. 3f). This implies that the ferritin moiety provides additional conformational stability to the ConM SOSIP.v7 trimers.

The ConM nanoparticles also interacted efficiently with the 19b and F105 non-NAbs at 4 °C and 37 °C, while the soluble trimers only did so at increased temperatures (>60 °C, Fig. 3f). The display of non-NAb epitopes implies that the ferritin nanoparticles also harbor non-native Env trimers, consistent with the presence of uncleaved ConM gp140 observed in sodium dodecyl sulfate-polyacrylamide gel electrophoresis (SDS-PAGE) analysis (Supplementary Fig. 5b) and the broader melting peak for ConM SOSIP.v7-ferritin nanoparticles compared to free ConM SOSIP.v7 trimers (Fig. 3e).

Next, we assessed whether presentation of ConM SOSIP.v7 trimers on ferritin nanoparticles provided an advantage in activation of B cells. B cell lines that can be induced to express bNAbs VRC01, PGT145, PGT121, or PG16 as their BCR[47] were incubated with soluble trimers or trimers presented on ferritin nanoparticles and the calcium flux upon encounter with antigen was measured by fluorescence-activated cell sorting analysis. At a concentration of 50 nM, the soluble trimers were able to weakly activate B cells expressing PGT121 and PG16 BCRs but were inactive against those expressing VRC01 and PGT145. In contrast, at the same molar amount of trimers (6.25 nM of nanoparticles), the ConM SOSIP.v7-ferritin nanoparticles activated all four B cells efficiently (Fig. 3g). These data imply that particulate presentation of ConM SOSIP.v7 trimers enhances their capability to activate cognate B cells.

**Immunogenicity of ConM SOSIP.v7 in rabbits.** To test the immunogenicity of ConM SOSIP.v7 trimers and ferritin nanoparticles, we vaccinated rabbits three times at weeks 0, 4, and 20 with 22 μg (Env protein equivalent) of either soluble ConM SOSIP.v7 trimers ($n = 5$) or ConM SOSIP.v7-ferritin nanoparticles ($n = 5$), formulated in ISCOMATRIX™ adjuvant (Fig. 4a)[3].

After two immunizations, all rabbits developed strong binding Ab responses against the corresponding ConM SOSIP.v7 trimer that waned but were boosted again by the third immunization (Fig. 4b). Four weeks after the prime, the binding Ab responses were seven-fold higher in the ConM SOSIP.v7-ferritin group compared to the single trimer ($p = 0.0159$), but these differences were less prominent (two-fold, $p = 0.0079$ at week 12) or non-significant (weeks 6, 20, and 22) after subsequent boosting immunizations (Fig. 4b).

To assess the induction of autologous NAb responses, we constructed an infectious molecular clone (IMC) containing the autologous ConM Env. This clone produced infectious virus at high titers that infected TZM-bl cells. Analysis of the sensitivity to a panel of bNAbs, non-NAbs, and reference serum pools revealed that the ConM virus was a highly neutralization-sensitive (Tier 1A) virus. However, the ConM virus was not sensitive to V3-directed non-NAbs, which is atypical for a Tier 1A virus (Supplementary Table 4)[48].

The autologous NAb responses mirrored the binding Ab responses (Fig. 4c). Thus, after two immunizations, all rabbits

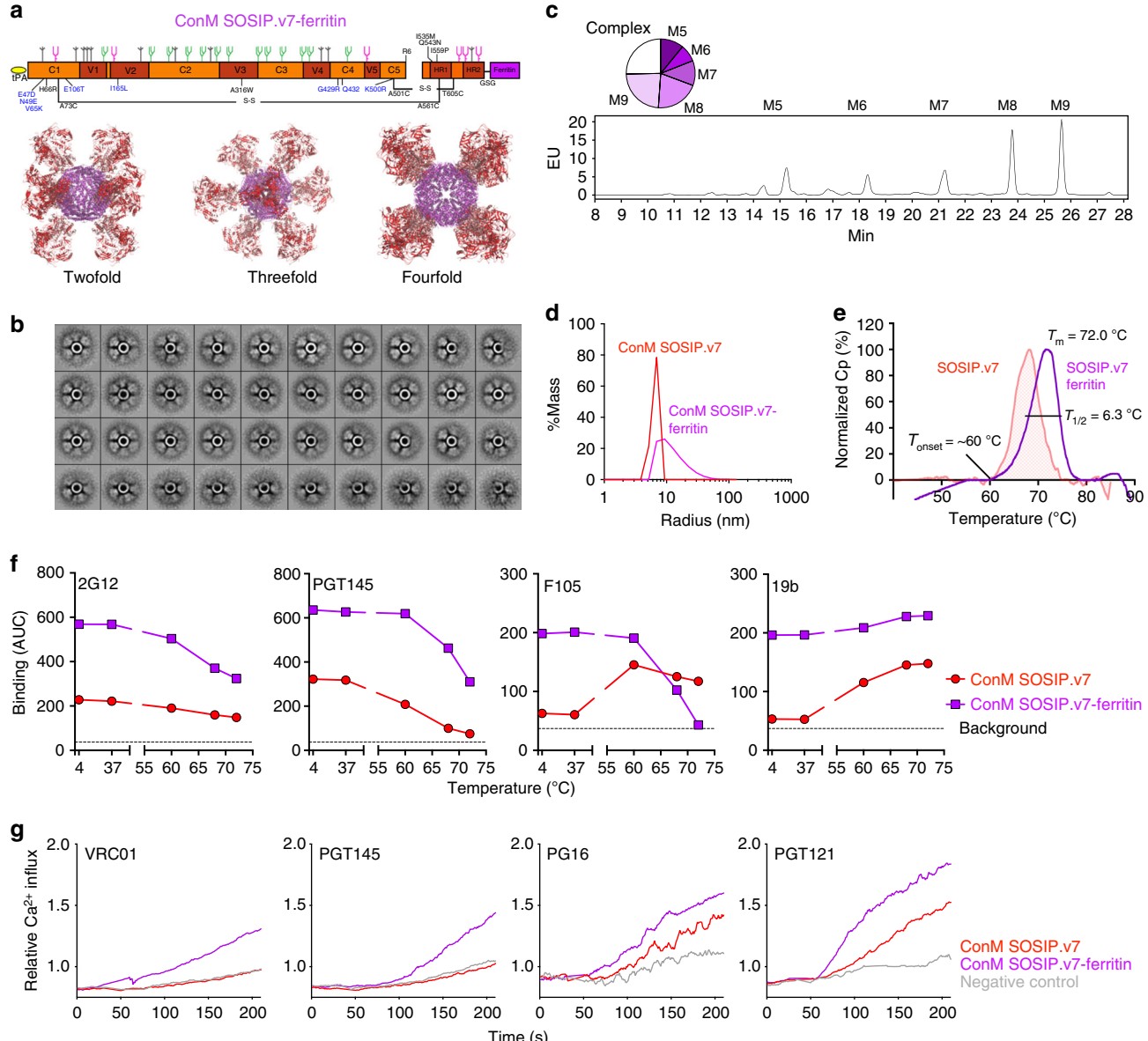

**Fig. 3** Design and characterization of ConM SOSIP-ferritin nanoparticles. **a** Left: linear representation of ConM SOSIP.v7-ferritin. The N-terminus from *H. pylori* ferritin (purple) is linked to the SOSIP.v7 C-terminus by a Gly-Ser-Gly (GSG) linker. Right: model of eight ConM SOSIP.v7 trimers with gp120 subunits (red) and gp41 subunits (pink), displayed on the *H. pylori* ferritin nanoparticle (purple, PDB: 3BVE) viewed down the two-fold, three-fold, and four-fold axis of the ferritin particle, respectively. The figure was drawn using PyMol. **b** The two-dimensional class averages of the negative-stain electron microscopic analyses of ConM SOSIP.v7-ferritin nanoparticle. Example images of the micrographs are shown in Supplementary Fig. 5c. **c** Representative dynamic light scattering graphs showing the size distribution of ConM SOSIP.v7 trimer and SOSIP.v7-ferritin nanoparticle. **d** Glycan profile of the ConM SOSIP.v7-ferritin nanoparticle as determined by hydrophilic interaction liquid chromatography-ultra performance liquid chromatography. The circle diagram indicates the relative amounts of $Man_{5-9}GlcNac_2$ (M5–M9) and complex glycans. **e** Thermostability (measured by differential scanning calorimetry) of PGT145-purified ConM SOSIP.v7-ferritin nanoparticle expressed in 293F cells. The ConM SOSIP.v7 trimer melting curve (Fig. 1e) is shown in red for comparison. **f** Binding of broadly neutralizing antibodies (bNAbs) 2G12 and PGT145 and non-neutralizing antibodies (non-NAbs) F105 and 19b to ConM SOSIP.v7 trimer and SOSIP.v7-ferritin (100 nM SOSIP.v7 equivalent each) measured by biolayer interferometry after incubation at different temperatures. Shown are the mean values of the area under the curve (AUC) derived from the binding curves in Supplementary Fig. 6. **g** Calcium flux (relative fluorescence) in B cells expressing either VRC01, PGT145, PGT121, or PG16 as the B cell receptor, stimulated with ConM SOSIP.v7 trimers or ConM SOSIP.v7-ferritin nanoparticles

developed strong autologous NAb responses against the corresponding ConM virus that waned and were boosted by the third immunization. The autologous NAb responses were significantly higher in the ConM SOSIP.v7-ferritin group compared to the single trimer group at week 4 (three-fold difference, $p = 0.0159$), week 6 (two-fold, $p = 0.0159$), and week 20 (four-fold, $p = 0.0159$) but not at week 22 (Fig. 4c). Furthermore, the ConM SOSIP.v7-binding Ab titers correlated

strongly with the autologous ConM NAb titers (Spearman $r = 0.88$, $p < 0.0001$; Fig. 4d).

We also tested neutralization against the related but more resistant ConS pseudovirus. The ConS virus was categorized as a Tier 1B virus, albeit at the resistant end of the Tier 1B spectrum. The ConS virus was completely resistant to most Abs that typically neutralize Tier 1 viruses, such as F105 and 447–52D (Supplementary Table 4). ConM represents the consensus of

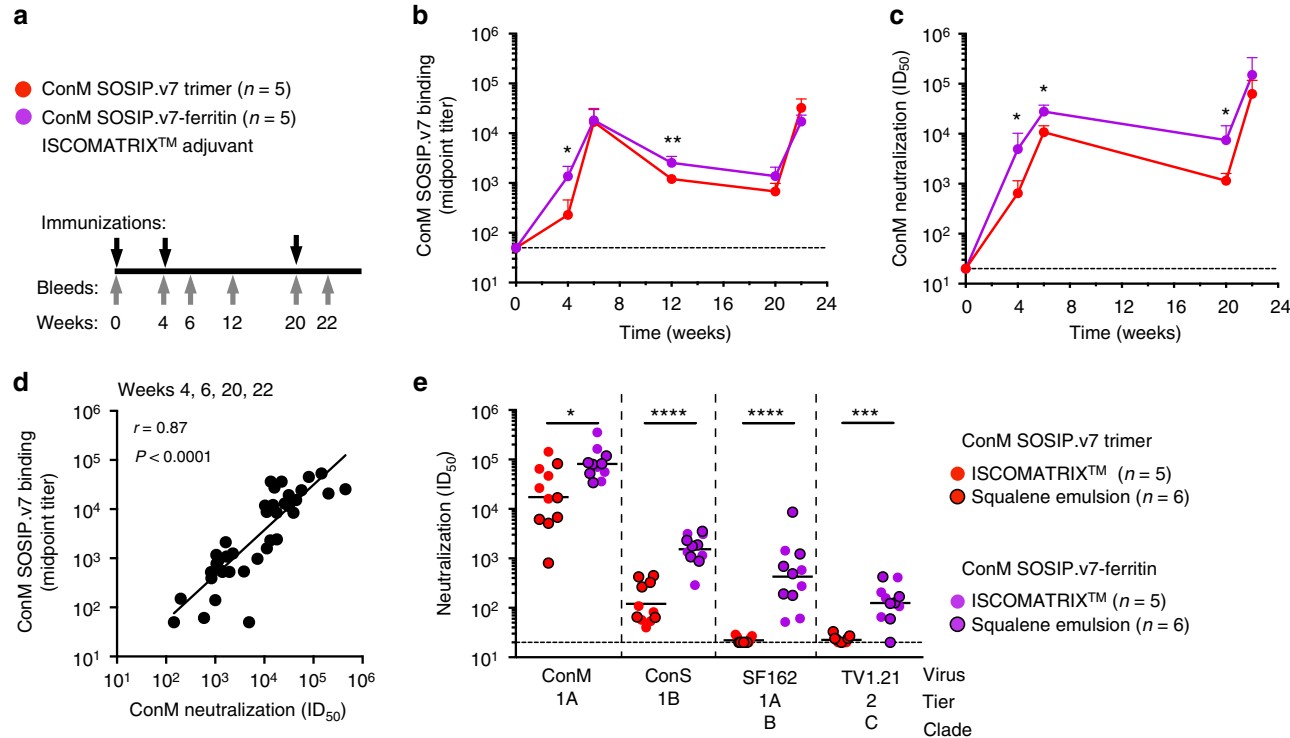

**Fig. 4** Immunogenicity of ConM trimers and nanoparticles in rabbits. **a** Rabbit immunization schedule. Rabbits were immunized at weeks 0, 4, and 20 (black arrows) and the antibody responses were analyzed at weeks 0, 4, 6, 12, 20, and 22 (gray arrows). **b** Midpoint antibody-binding titers over time against ConM SOSIP.v7 trimer as measured by D7324-capture enzyme-linked immunosorbent assay. The dotted line indicates a midpoint titer of 50, the lower cut-off of the assay. The mean binding titers with SD ($n = 5$ animals) of each group are shown. Two-tailed Mann–Whitney $U$ test was used to determine statistical differences between groups at each time point. *$p < 0.05$; **$p < 0.01$. **c** Neutralization titers ($ID_{50}$) over time against the infectious autologous ConM virus as measured in the TZM-bl assay. The dotted line indicates a midpoint titer of 20, the lower cut-off of the assay. The mean neutralization titers with SD ($n = 5$ animals) of each group are shown. Two-tailed Mann–Whitney $U$ test was used to determine statistical differences between groups at each time point. *$p < 0.05$. **d** Antibody binding titers against ConM SOSIP.v7 trimer and autologous neutralization titers were cross-compared over all post-prime time points. The Spearman $r$ and $p$ values are given. **e** Neutralization titers ($ID_{50}$) from rabbit sera samples at week 22 (from two different immunization studies) against autologous ConM virus (DUMC and AMC), heterologous Tier 1 SF162 and MW965.26 viruses (AMC and DUMC), and heterologous Tier 1B ConS and Tier 2 TV1.21 virus (DUMC and AMC). Neutralization data from individual animals and the different test laboratories can be found in Supplementary Table 5. Neutralization titers of the sera from rabbits that were immunized with the same immunogens but with a different adjuvant are indicated with or without black bordered symbols in the figure. Horizontal lines indicate the geometric mean values. A two-tailed Mann–Whitney $U$ test was used to compare differences; *$p < 0.05$; ***$p < 0.001$; ****$p < 0.0001$

consensus sequences from 2004 (i.e., the last year consensus sequences were reported at the Los Alamos website), while ConS is based on a consensus sequence from 2001 and has longer variable loops[12]. In total, the ConM Env is 23 amino acids shorter than ConS Env with the V1, V2, V4, and V5 loops of ConM being 3, 4, 8, and 3 amino acids fewer than those of ConS, respectively. In addition, the ConS sequence has 27 amino acid changes compared to ConM scattered throughout the gp160 sequence (Supplementary Fig. 1). In this analysis, we included 12 additional week 22 rabbit sera from an independent immunization study in which rabbits received 20 µg ConM SOSIP.v7 or ConM SOSIP.v7-ferritin ($n = 6$ animals per group), also at weeks 0, 4, and 20, but formulated in squalene emulsion (SE) as the adjuvant[49].

At week 22, 21 out of 22 rabbits had developed NAb activity ($ID_{50}$ titer >40) against the ConS pseudovirus and the levels were ~22-fold higher in the nanoparticle recipient animals (median $ID_{50}$ of 1771 for the nanoparticle group versus 82 for the single trimer group, $p < 0.0001$; Fig. 4e). The sera were assayed in 2–4 different laboratories and all gave similar results (Supplementary Table 5). Thus, particulate presentation of ConM SOSIP.v7 trimers was required to induce a strong response against the related ConS virus. Analyses of the longitudinal autologous neutralization responses (Fig. 4c) and ConS neutralization

responses (Fig. 4e) revealed that post-immunization 1 (week 4), post-immunization 2 (week 6 and week 20), and post-immunization 3 (week 22) Tier 1A ConM NAb responses correlated with post-immunization 3 (week 22) ConS NAb responses (Supplementary Fig. 7). The correlation was particularly strong for week 6 ConM NAb responses versus the week 22 ConS NAb responses (Spearman $r = 0.9167$, $p = 0.0013$). Thus, early Tier 1A ConM NAb responses were predictive for later ConS NAb responses.

Heterologous NAb responses against the Tier 1A clade B virus SF162 were weak in the trimer group, consistent with the presence of substitutions in the SOSIP.v7 design that limit exposure of non-neutralizing epitopes[5,16,17] but stronger in the nanoparticle group (median $ID_{50}$ of 20 versus 488, $p < 0.0001$) (Fig. 4e). Similarly, the Tier 1A clade C virus MW965.26 was neutralized more efficiently by nanoparticle recipients than by trimer-immunized animals (median $ID_{50}$ of 9982 versus 116, $p = 0.0079$) (Supplementary Table 5). The heterologous Tier 1A NAb activity in most nanoparticle-immunized animals was partially depleted by the ConM V3 peptide, while the Tier 1A NAb activity for most trimer immunized animals was barely affected (Supplementary Table 7). Thus, the exposed V3 on ConM SOSIP.v7-ferritin (Fig. 3f)

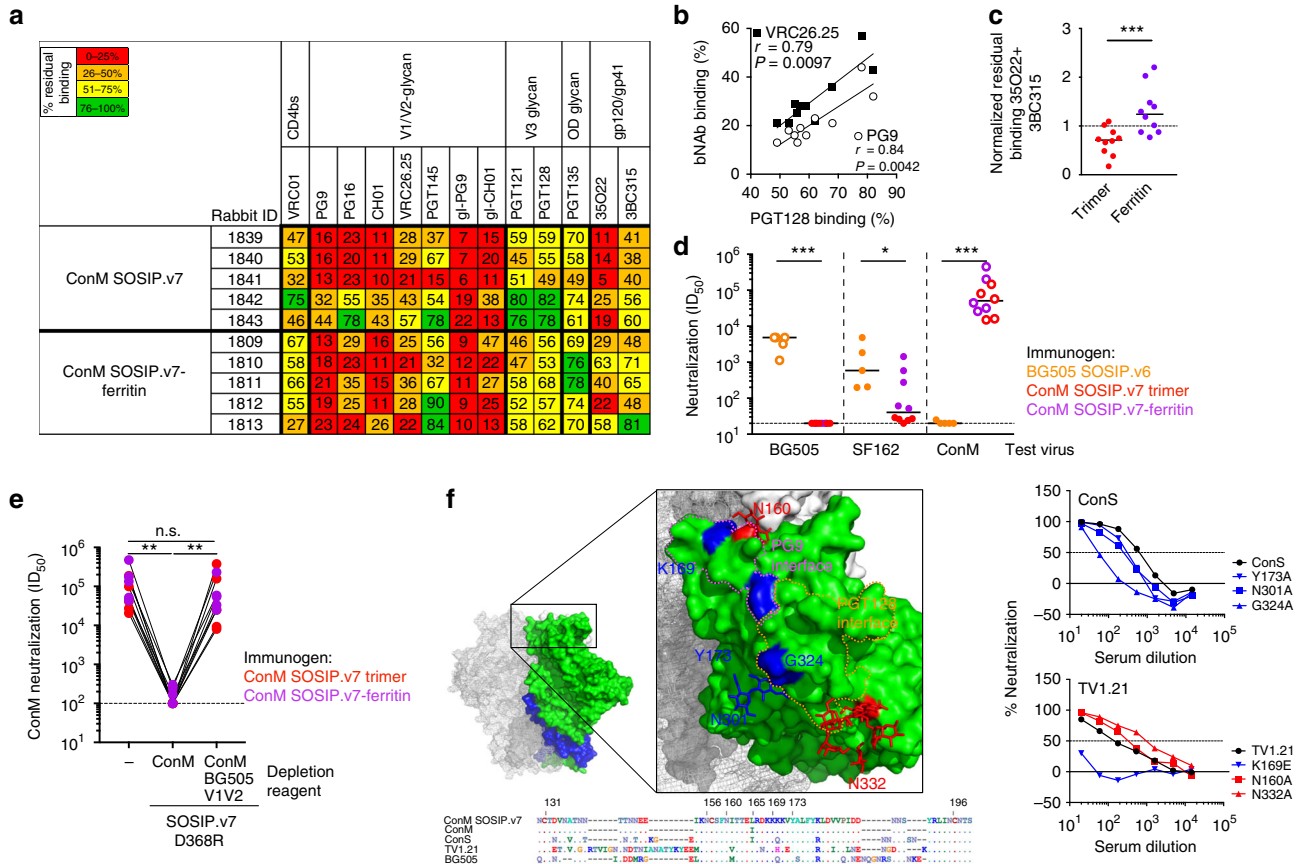

**Fig. 5** Mapping of antibody specificities induced by the ConM SOSIP.v7 trimer in rabbits. **a** Enzyme-linked immunosorbent assay-based competition assay measuring the interference of rabbit sera with the binding of known broadly neutralizing antibodies (bNAbs) to the ConM SOSIP.v7 trimer. The numbers represent the residual bNAb binding (0–25% in red; 26–50% in orange; 51–75% in yellow; 76–100% in green) after pre-incubation of the ConM SOSIP trimer with a 1:100 dilution of the rabbit sera. The recorded values are average values from four independent experiments. **b** Correlation between serum competition with PGT128 and serum competition with PG9 and VRC26.25 as determined in Fig. 5a. The Spearman $r$ and $p$ values of the correlations are given. A complete Spearman correlation matrix is shown in Supplementary Fig. 8. **c** Normalized residual binding of 3BC315 and 35O22 to ConM SOSIP.v7 trimer after adding sera of ConM SOSIP.v7 and ConM SOSIP.v7-ferritin immunized rabbits. Data are derived from Fig. 5a. Differences were determined by two-tailed Mann–Whitney $U$ test; ***$p < 0.001$. **d** ConM NAb responses in week 22 sera from rabbits immunized with ConM SOSIP.v7 trimer (5 animals) or ConM SOSIP.v7-ferritin nanoparticle (5 animals) compared to historic control sera (week 22) from rabbits that were immunized with BG505 SOSIP.v6 (5 animals; animal ID: 1829–1833)[5]. Neutralization activity was assessed against BG505, SF162, and ConM viruses. Open symbols indicate when the autologous virus was tested. A two-tailed Mann–Whitney $U$ test was used to compare differences; *$p < 0.05$; ***$p < 0.001$. **e** Neutralization titers against ConM in the presence or absence of ConM SOSIP.v7 D368R or a ConM SOSIP.v7 D368R in which the V1V2 was swapped for that of BG505. Statistical analysis was performed using the Wilcoxon matched-pairs signed-rank test. **$p < 0.01$. **f** Conformational epitopes in the V2–V3 region of ConM SOSIP.v7 that elicit strong NAbs. Residues within the apex and N332_V3 conformational epitopes are colored in blue (K169, Y173, N301, and G324) and red (Asn160 and Asn332) with glycans represented by stick models. Interfaces that are recognized by bNAbs PG9 and PGT128 are marked with dotted lines in magenta and orange, respectively. Neutralization curves of one rabbit serum (#1811, Supplementary Table 7) against key mutant viruses on the right. Bottom: alignment comparing the V1V2 amino acid sequences of ConM SOSIP.v7 with that of the ConM, ConS, TV1.21, and BG505 viruses

induced anti-V3 Tier 1A NAb responses in rabbits, while the Tier 1A neutralization from soluble trimer-immunized animals is probably mediated by non-V3 specificities.

Instances of heterologous Tier 2 virus neutralization at a titer >40 were virtually absent from the single trimer group. Such hits, although weak and usually sporadic, were more frequent in the nanoparticle group, but this finding was driven mostly by NAb responses against the clade C virus TV1.21, which was not neutralized by any sera from the single trimer recipients but by 10 out of 11 sera from the nanoparticle group (median ID50 of 20 versus 127, $p < 0.0002$ for the comparison; Fig. 4e; Supplementary Table 5).

**NAbs induced by ConM SOSIP.v7 target the trimer apex.** To probe the specificities of the Ab responses, we performed serum-binding competition experiments with bNAbs[3,50]. The sera were

able to compete with many bNAbs for binding to the trimer (Fig. 5a). Particularly strong competition was observed for several bNAbs targeting the V1V2 region on the trimer apex, such as PG9, PG16, CH01 and VRC26.25 (<40% residual binding of these bNAbs in the presence of sera from 8 out of 10 rabbits), except for PGT145 (Fig. 5a). Very strong competition was also detected for gl-PG9 and gl-CH01, showing that ConM immunogens are not only able to interact with gl-bNAbs but also induce Abs that interact with overlapping epitopes. Moderate competition was also observed with VRC01 and for some sera with the two anti-V3 glycan bNAbs. The competition of V3-glycan bNAbs (e.g., PGT121 and PGT128) and V1V2 bNAbs (e.g., VRC26.25 and PG9) correlated well (Fig. 5b and Supplementary Fig. 8), suggesting that ConM Abs were targeting epitopes near or over-lapping with the V3-glycan and V1V2 regions. Furthermore, we found that sera from nanoparticle-immunized rabbits showed

weaker competition with the 35O22 and 3BC315 bNAbs targeting the gp41–gp120 interface[40,51] (both bNAbs combined, $p = 0.001$; Fig. 5c). This might be explained by poor accessibility of the gp41–gp120 interface when the trimers are presented on ferritin nanoparticles[44]. In contrast, soluble ConM trimers expose a glycan-free area on their base that induces Abs that can compete with 3BC315 and 35O22[52].

Next, we attempted to determine the specificity of the NAb response against the autologous ConM virus. The Tier 1A status of this virus prompted us to investigate whether there was a role for V3, despite the fact that ConM is not very sensitive to most V3 non-NAbs (Supplementary Table 4). NAbs that target linear V3 epitopes usually do not contribute to Tier 2 NAb responses and are therefore not useful on the path toward a bNAb-based vaccine[3]. We used a cyclic ConM V3 peptide to deplete the autologous ConM NAbs from the sera of the ten rabbits. The V3 peptide was unable to deplete the ConM NAb responses from any of the sera, while the ConM SOSIP.v7 D368R protein depleted almost all neutralization activity (Supplementary Table 6).

To confirm that simple linear V3 specificities played no significant role in autologous NAb responses, we made use of five historic control sera from animals that were vaccinated with BG505 SOSIP.v6 (animal IDs: 1829–1833; for the BG505 SOSIP.v6 sequence, see Supplementary Fig. 1) and neutralized the Tier 1A virus SF162, through V3 specificities[5] (Fig. 5d). ConM and BG505 have identical V3 loop sequences, but despite the sequence similarities in the V3 and the presence of V3-directed Tier 1A NAbs, none of the historic control sera neutralized the ConM virus, confirming that the autologous ConM response is not a typical V3-directed Tier 1A NAb response.

The competition ELISA suggested the V1V2 as a major target for binding Abs and the same Abs might also be responsible for neutralizing the autologous virus. To investigate, we generated a ConM SOSIP.v7 D368R protein in which the V1V2 domain was replaced with that of BG505. This hybrid trimer was purified using PGT145 and used in neutralization-depletion experiments. While the parental ConM SOSIP.v7 D368R trimer depleted all NAb activity from the sera of all ten rabbits, the hybrid trimer with the BG505 V1V2 domain had no such effect, showing that the ConM V1V2 domain is (part of) the dominant ConM NAb response in all rabbits (Fig. 5e). Together, these results suggest that NAbs against the autologous Tier 1A ConM virus do not target linear V3 epitopes but epitopes on or very near the V1V2 on the trimer apex.

Next, we evaluated the NAb activity of ten rabbit sera against a panel of ConS virus mutants (Supplementary Table 7). Substitutions Y173A, N301A (removing a glycan), and G324A consistently reduced ConS sensitivity to NAb activity, providing almost complete resistance against the NAb activity in the five sera from the trimer-recipient animals, and considerably reducing sensitivity to the five sera from nanoparticle recipients. The G324A substitution had the largest effect (up to 30-fold reduction in sensitivity; Supplementary Table 7). In contrast, a N332A substitution, removing the glycan at that position, consistently enhanced NAb sensitivity, although the effect was subtle.

The sera that neutralized the TV1.21 virus, i.e., the sera from the five nanoparticle recipients, were also tested against a small panel of TV1.21 mutants (Supplementary Table 7). The K169E substitution completely abrogated TV1.21 neutralization, while the N160K and N332A substitutions consistently enhanced TV1.21 sensitivity to the five sera by approximately two-fold and four-fold, respectively (Supplementary Table 7). Combined, these data suggest that the NAb specificities against the ConS and TV1.21 viruses are directed against conformational epitopes around the V3 base and apex involving residues 169 and 173 in V2, residue 324 in V3, and the glycan at position 301 in V3. These

residues are in close proximity to the epitopes for PG9 and PGT128, in line with the strong serum competition with these bNAbs (summarized in Fig. 5f). Removal of the glycans at positions 160 and 332 glycan appears to enhance exposure of this epitope.

In summary, ConM SOSIP.v7 trimers, especially when presented on nanoparticles, induce NAbs that target the apex around the V1V2 and V3 regions on the ConM, ConS, and TV1.21 viruses, and the sera of most rabbits competed with the binding of (gl-)bNAbs that target this region, suggesting that these regions form the immunodominant epitope(s) of ConM SOSIP.v7 in rabbits.

**ConM SOSIP.v7 nanoparticles induce ConS NAbs in macaques**. We next evaluated the immunogenicity of ConM SOSIP.v7-ferritin nanoparticles in rhesus macaques when combined in a trivalent cocktail with AMC008 SOSIP.v5.2-ferritin and AMC011 SOSIP.v5.2-ferritin nanoparticles[5,16,44,53]. Six animals received three intramuscular vaccinations at weeks 0, 4, and 20 of 33 μg of each nanoparticle construct (i.e., 100 μg nanoparticle protein mass in total per animal per time point) formulated in AddaVax adjuvant (Fig. 6a). Two control animals received adjuvant only and were negative for all specific serological assays throughout the experiment.

After two immunizations with the nanoparticle cocktail, the rhesus macaques developed strong binding responses against the ConM SOSIP.v7 trimer that waned and were then boosted by the third vaccination (Fig. 6b), which mirrored the results of the rabbit immunizations (Fig. 4b). Furthermore, all vaccinated macaques induced specific Ab responses against the ferritin component. Similar binding Ab responses were also detected in the rabbits that received ferritin nanoparticles (Supplementary Fig. 9), indicating that *H. pylori* ferritin is immunogenic in both animal models, consistent with previous findings using ferritin nanoparticles displaying influenza hemagglutinin immunogens[54].

The ConM NAb activity measured over time mirrored the ConM SOSIP.v7-binding responses and peaked at weeks 6 and 22 (Fig. 6c). The sera from all animals that received the nanoparticle cocktail vaccine neutralized the autologous ConM virus with a median $ID_{50}$ of 1880 at week 22. The ConM NAb activity in the macaque sera was depleted by the autologous trimer but not by the hybrid trimer containing the BG505 V1V2 domain nor by the autologous V3 peptide (Fig. 6d), suggesting that the macaque ConM NAbs targeted epitopes that were shared with the rabbit NAbs. The related heterologous ConS virus was neutralized by all six animals, albeit with ~15-fold lower titers compared to the ConM virus (median $ID_{50}$ of 123) (Fig. 6e).

Neutralization experiments with the ConS mutant viruses yielded somewhat different results compared to those obtained with the rabbit sera. For example, viruses with N332A and T607S substitutions were neutralized less efficiently by macaque sera, suggesting that the ConS NAbs in the macaques target somewhat different epitopes than those in the rabbits and/or that these epitopes are approached differently (Fig. 6f, Supplementary Table 9). The sera of the immunized macaques also neutralized the heterologous Tier 1A SF162 and MW965.26 viruses (median $ID_{50}$ values of 837 and 739, respectively), and these NAbs were partially depleted by the V3 peptide (Fig. 6e, Supplementary Table 9). The AMC008 and AMC011 nanoparticles in the cocktail vaccine did not induce autologous NAb responses in these animals nor did we detect neutralization against heterologous Tier 2 viruses, including TV1.21 (Fig. 6e and Supplementary Table 8). In summary, ConM SOSIP.v7-ferritin nanoparticles are able to induce NAbs against the autologous Tier 1A ConM virus and the related ConS virus in macaques.

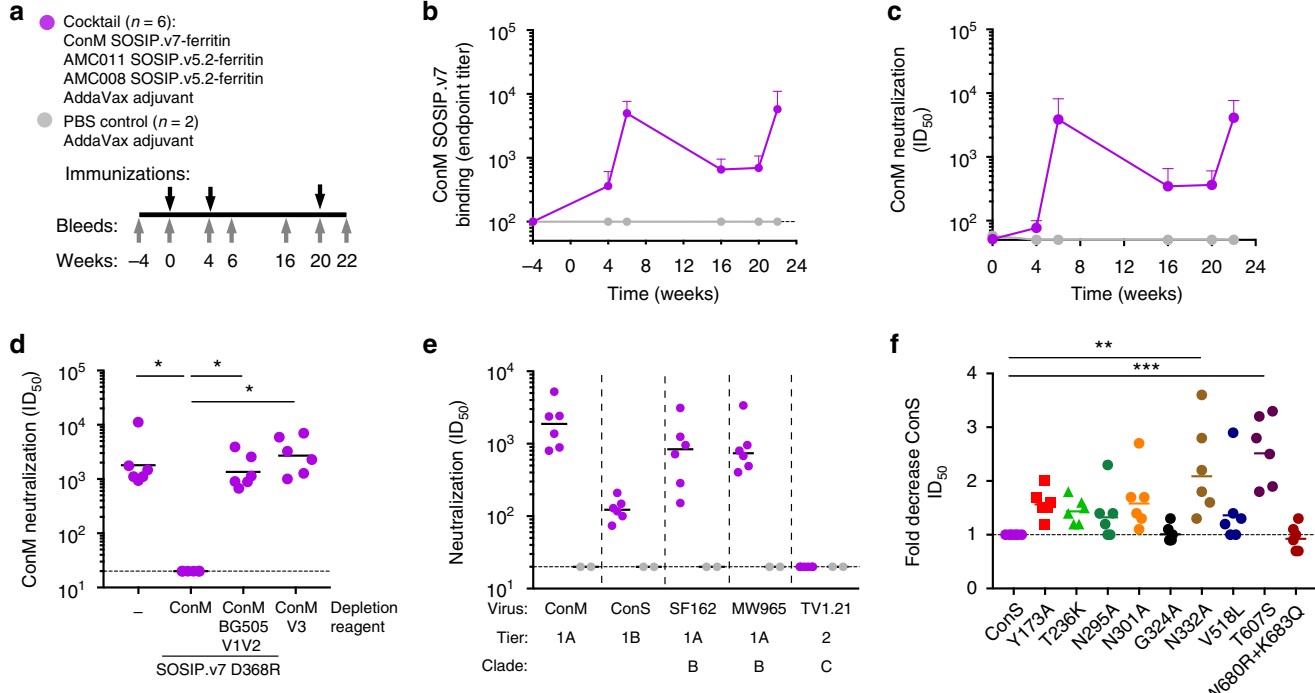

**Fig. 6** Immunogenicity of ConM SOSIP.v7-ferritin in macaques. **a** Macaque immunization schedule. Macaques were immunized at weeks 0, 4, and 20 (black arrows) and the Ab responses were analyzed at weeks −4, 0, 4, 6, 16, 20, and 22 (gray arrows). **b** Endpoint antibody-binding titers over time against ConM SOSIP.v7 trimer as measured by D7324-capture enzyme-linked immunosorbent assay. Mean titers ($n = 6$ animals) with SD are shown. Values are from one experiment performed in duplo. **c** Neutralization titers with over time against ConM virus. Mean titers ($n = 6$ animals) with SD are shown. **d** Neutralization titers against ConM in the presence or absence of ConM SOSIP.v7 D368R, ConM SOSIP.v7 D368R with the V1V2 of BG505, or the ConM V3 peptide. Statistical analysis was performed using the Wilcoxon matched-pairs signed-rank test; *$p < 0.05$. **e** Neutralization titers ($ID_{50}$) from macaque sera samples at week 22 against autologous ConM virus (DUMC), heterologous Tier 1 SF162 virus (AMC), and MW965.26 virus (DUMC), and heterologous ConS and TV1.21 virus (DUMC). Neutralization data from individual animals can be found in Supplementary Table 8. **f** Relative neutralization titers of sera against a panel of ConS mutants. Neutralization data from individual animals can be found in Supplementary Table 9. Statistical analysis was performed by Kruskal–Wallis test and Dunn's test to correct for multiple testing. **$p < 0.01$ and ***$p < 0.001$.

## Discussion

We described the design, characterization, crystal structure, and immunogenicity of ConM SOSIP.v7, a native-like trimer based on the consensus sequence of HIV-1 group M. ConM SOSIP.v7 can be purified in high yields, has all of the hallmarks of a properly folded native-like trimer, and binds to several inferred germline bNAbs. The crystal structure described here provides a useful starting point for designing future immunogens. Although we did not perform a direct comparison with ConM gp120 or nonnative ConM gp140 forms, the ConM SOSIP.v7 trimer induced very strong autologous NAb responses against the autologous Tier 1A ConM virus in animals, especially when presented on ferritin nanoparticles, and induced substantial NAb responses against the related ConS virus.

Native-like Env trimers described thus far were based on Env sequences from single natural virus isolates[55–57] with few exceptions[37,58]. Here we showed that it is possible to generate stable native-like Env trimers from an in silico derived, i.e., "non-natural," Env sequence. The design of this consensus trimer was facilitated by combining the original SOSIP.664 mutations[4,6,7,59,60] with the more recent H66R, A316W, I535M, and Q543N substitutions (SOSIP.v4.2[16,61,62]), the A73C-A561C intersubunit disulfide bond (SOSIP.v5.2[5]), and additional substitutions described by Guenaga et al.[17]. This combination of modifications, here termed SOSIP.v7, might constitute a general recipe for making native-like trimers from a given Env sequence, natural or unnatural, and thus a next-generation SOSIP trimer design platform.

The ConM SOSIP was made to minimize clade-specific or individual isolate-specific antigenic determinants with the goal of generating a component of a bNAb-inducing vaccine. Indeed, the ConM trimer contains less rare surface-exposed residues than BG505 and thus probably displays less strain-specific antigenic determinants. Fewer ConM-specific antigenic determinants might generate less potentially distracting specific Abs and facilitate driving increased neutralization breadth. By determining the ConM-specific antigenic determinants, ConM SOSIP.v7 could be further engineered to reduce ConM-specific rare residues on the surface while maintaining its overall intact trimeric structure and antigenicity for bNAbs.

The mapping of the NAb responses point to involvement of epitopes at or near the V1V2 at the trimer apex. The immuno-dominant epitope(s) on ConM trimers is/are thus quite different, compared with those on BG505 SOSIP trimers that often involve a hole in the glycan shield on trimer stem, and might make the response to the ConM trimer more amenable for broadening[10,63,64]. This notion is supported by the observation that ConM trimers induce Abs that compete with (gl-)bNAbs that target the trimer apex. Such apex responses are much less likely to be induced by previously generated group M consensus immunogens based on, for example, gp120 or non-native trimers[2,12–15,65], because the trimer apex is likely not appropriately presented on these immunogens. Isolation and characterization of monoclonal NAbs from the ConM immunized animals will be necessary to determine the precise mode of

binding of NAbs that target ConM, ConS, and TV1.21 and this will be crucial to inform future vaccine design improvements.

The induction of NAb responses by ConM SOSIP.v7 trimers benefited from presentation on ferritin nanoparticles, as the rabbits in the nanoparticle groups developed enhanced autologous Tier 1A ConM NAb responses, as well as Tier 1B ConS and Tier 2 TV1.21 responses. The beneficial effects of particulate antigen presentation through more efficient B cell activation are well known[43,44,54,66,67]. Indeed, using calcium flux assays, we confirmed that ConM SOSIP.v7-ferritin nanoparticles strongly activated B cells bearing bNAbs PG16, PT145, VRC01, or PGT121 as the BCR.

There is still, however, room for improvement of the nanoparticle platform. First, the ferritin component induced anti-ferritin Ab responses, which might distract from Env-specific bNAb responses. A solution could be to immunosilence ferritin by adding glycans[68]. Second, the NS-EM and DLS analyses showed that, in addition to fully assembled particles displaying eight trimers, incomplete particles were also present. Third, a portion of the trimers on the nanoparticles was not cleaved at the gp120/ gp41 junction as assessed by SDS-PAGE analysis (Supplementary Fig. 5b). In general, one-component nanoparticles that assemble intracellularly, such as ferritin, might be suboptimal because it will be difficult to ensure that all trimers on each particle are bona fide native-like trimers. Novel nanoparticle platforms, in particular two-component nanoparticle systems, in which one can first purify single native-like trimers and then assemble the nanoparticles post-purification, offer opportunities to do so[69].

Following the same overall consensus sequence strategy, one could pursue a similar approach for generating Env immunogens that are based on the consensus sequence of the majority circulating sequences in a specific geographical region. Therefore, we have recently generated Env immunogens using consensus sequences of clades B, C, and H using the same SOSIP.v7 design (unpublished), which are currently being explored further and could be instrumental reagents for regional vaccine strategies. Rutten et al. have also described a SOSIP trimer based on the consensus sequence of clade C[37]. Immunogens based on a consensus sequence could be very useful for other pathogens with high diversity, such as influenza virus[70], Ebola virus, and hepatitis C virus[71]. The observation that the consensus sequences can form properly folded native-like Env proteins in high yield shows that such strategies are feasible.

Although ConM SOSIP.v7 trimers are unlikely to become stand-alone immunogens for inducing bNAbs, they may be very useful in an immunization regimen. For example, they could be suitable immunogens to boost responses that are primed by a germline targeting trimer[29,38]. The ConM trimer is able to bind to the inferred germline precursors of several bNAbs against the trimer apex and can induce Abs that compete with these same gl-bNAbs, which is highly relevant in this context. The low abundance of strain-specific antigenic determinants is an advantage. Thus, we propose that the ConM SOSIP.v7 trimer is a suitable component in immunization regimens aimed at inducing bNAbs. A cGMP stock of ConM SOSIP.v7 has recently been produced and the protein will be evaluated in phase I clinical trials starting in 2019.

## Methods

**Construct design**. The ConM *env* gene is derived from the last published group M consensus sequence (2004; www.hiv.lanl.gov; alignment ID: 104CP2, Supplementary Fig. 1), which is a consensus sequence of all of the consensus sequences of each clade in group M. The ConM SOSIP.v4.2 construct was designed as previously described[4,16]. To improve the formation of soluble trimers, the construct contained the following changes compared to the original ConM *env* sequence: a TPA signal sequence, A501C and T605C (gp120-gp41$_{ECTO}$ disulfide bond), I559P in gp41$_{ECTO}$ (trimer-stabilizing), H66R and A316W (trimer-stabilizing), I535M and Q543N in

gp41$_{ECTO}$ (improved trimerization), REKR to RRRRRR (R6) in gp120 (cleavage enhancement), and a stop codon after gp41$_{ECTO}$ residue 664. The construct was codon-optimized, ordered from Genscript (Piscataway, NJ, USA), and cloned into the pPPI4 plasmid. A second intermolecular disulfide bond was then added by introducing mutations A73C in gp120 and A561C in gp41[5], resulting in ConM SOSIP.v5.2. Addition of TD8 mutations (E47D, N49E, V65K, E106T, I165L, G429R, and K500R; the sequence already contains a Q at position 432)[17] resulted in ConM SOSIP.v7 (Supplementary Table 1). The interprotomer disulfide bond that is part of the SOSIP.v6 design was not included in the SOSIP.v7. Point mutations were made by Quickchange site-directed mutagenesis (Agilent Technologies, La Jolla, CA, USA) and verified by sequencing. Untagged trimers were used for experiments described in Figs. 1c and 3f and for all immunizations. In addition to untagged trimers, we also generated D7324-tagged and His-tagged trimers[4]. D7324-tagged constructs were generated by adding the amino acid sequence GSAPTKAKRRVVQREKR after residue 664 in gp41$_{ECTO}$. His-tagged constructs were generated by adding the amino acid sequence GSGSGGGSGHHHHHHHHH after residue 664. The GS amino acid sequence (underlined) in both tags contains a BamHI restriction site. D7324-tagged trimers were used for experiments described in Figs. 1d–f and 3c and Supplementary Figs. 2a–d and 5a, while His-tagged trimers were used for the experiments described in Fig. 1g. The ConM SOSIPv7-ferritin construct was generated by fusing the N-terminus from *H. pylori* ferritin (Genbank accession no. NP_223316), starting from Asp5, to the SOSIP.664 C-terminus, separated by a GSG linker[44]. The plasmid containing ferritin only was constructed by replacing the ConM SOSIP. v7 sequence in the ConM SOSIP.v7-ferritin plasmid with a N-terminal His6-tag followed by a GGSG-linker.

The IMC of ConM was constructed by using the molecular clone of LAI as backbone (pLAI)[73]. This clone contains a unique SalI restriction site 434 nucleotides upstream of the *env* start codon and a unique BamHI site at the codons specifying amino acids G751 and S752 in LAI gp160 (HxB2 numbering). A DNA fragment containing the LAI sequences between the SalI site and the *env* start codon, followed by the ConM *env* sequence up to the BamHI site, was synthesized and cloned into the LAI molecular clone backbone using SalI and BamHI (Genscript, Piscataway, NJ). The resulting molecular clone encodes the complete ConM gp160 sequence, except for the C-terminal 106 amino acids of the cytoplasmic tail, which are derived from LAI gp160.

**Protein expression and purification**. Adherent HEK293T cells (ATCC, CRL-11268) were maintained in Dulbecco's Modified Eagle's Medium (DMEM) supplemented with 10% fetal calf serum (FCS), penicillin (100 U/mL), and streptomycin (100 μg/mL) and transfected as described previously[16]. The cell supernatants containing unpurified SOSIP trimers were harvested 3 days after transfection. Suspension 293F (Invitrogen, cat no. R79009) and 293S cells (ATCC, CRL-3022) were maintained in FreeStyle medium (Life Technologies) and co-transfected using 1 mg/mL PEImax (Polysciences Europe GmBH, Eppelheim, Germany) at a density of 0.8–1.2 million cells/mL with a plasmid expressing SOSIP and a plasmid encoding furin in a 4:1 ratio to ensure gp140 cleavage or 1:1 when expressing ConM SOSIP.v7-ferritin nanoparticles[44]. The supernatant was harvested 7 days after transfection, centrifuged, and filtered using Steritops (0.22 μm pore size; Millipore, Amsterdam, The Netherlands) before further use. SOSIP trimers were purified essentially as previously described[16,72]. In brief, filtered supernatant from 293F or 293S cells was flowed (0.5–1.0 mL/min) over a PGT145 or 2G12 affinity column, prepared as previously described[72], at 4 °C. 2G12 purification was followed by SEC. Protein concentrations were determined with Nanodrop (Thermo Scientific, Wilmington DE, USA) using theoretical molecular weight and extinction coefficient calculated with Expasy (ProtParam tool).

The ConM SOSIP.v7-ferritin nanoparticles were purified by adding the CNBr-activated sepharose 4B beads (GE Healthcare) carrying PGT145 to the filtered supernatant and incubated on a roller at 4 °C overnight. Subsequently, the supernatant and beads were passed over an Econo-Column chromatography column (Biorad). The column was then washed with three column volumes of 0.5 M NaCl and 20 mM Tris HCl pH 8.0. Protein was eluted with 3.0 M MgCl$_2$ pH 7.5 and immediately buffer exchanged into TN75 buffer (75 mM NaCl, 20 mM Tris HCl pH 8.0) using a 100-kDa cut-off Vivaspin20 filter (Sartorius, Göttingen, Germany).

For crystallography, the PGT124 and 35O22 Fabs were overexpressed by transient transfection in the FreeStyle 293F cells for 5 days and purified by affinity chromatography using CaptureSelect LC lambda (Thermo Fisher Scientific), cation exchange chromatography, and SEC with Superdex200 16/60 column (GE Healthcare). The ConM SOSIP.v7 construct was cloned into a phCMV3 vector and overexpressed in FreeStyle 293S cells for 6 days and purified by 2G12-affinity chromatography, followed by SEC. The purity of all trimers and Fabs was assessed using SDS-PAGE and Blue Native-PAGE (BN-PAGE), followed by Coomassie blue staining as previously described[4]. The His$_6$-tagged ferritin nanoparticles (without Env trimers) were expressed in FreeStyle 293F cells and purified by gravity flow over a Ni-NTA column (Qiagen) followed by SEC over a Superdex200 10/300 GL increase column. Fractions corresponding to the size of the ferritin 24-mer were pooled and concentrated in phosphate-buffered saline (PBS).

**SDS-PAGE and BN-PAGE**. SOSIP trimers were analyzed using SDS-PAGE and BN-PAGE followed by western blotting or Coomassie blue staining as described previously[16].

**Enzyme-linked immunosorbent assay**. For D7324-capture ELISA, supernatants from HEK293T cells containing unpurified SOSIP trimers, or PGT145-purified SOSIP trimers (1.0 μg/mL), were diluted in Tris-buffered saline (TBS). The trimers were then immobilized via their D7324-tags for 2 h at room temperature on half-well 96-well plates (Greiner) precoated with Ab D7324 (Aalto Bioreagents) at 10 μg/mL in 0.1 M NaHCO₃ pH 8.6 overnight[4,50]. Midpoint and end point antibody titers from sera samples were determined in D7324-capture ELISA as previously described[3]. For lectin-capture ELISA, half-well 96-well plates were coated with *Galanthus nivalis* lectin (Vector Laboratories) at 20 μg/mL in 0.1 M NaHCO₃ pH 8.6 overnight, which were then blocked using Casein Blocker (Thermo Scientific) followed by immobilization of the purified SOSIP trimer and SOSIP-ferritin proteins in TBS (2.0 μg/mL). Subsequent steps were performed as previously described[4]. For Ni-NTA His-capture ELISA, the purified SOSIP trimers (1.0 μg/mL) were diluted in TBS and immobilized on 96-well Ni-NTA ELISA plates (Qiagen). Subsequent steps were performed identical to the D7324-capture ELISA. For the ELISA in Supplementary Fig. 9, 2 μg/mL of purified ferritin cages were coated overnight on half-well 96-well plates, which were then blocked using Casein Blocker. Subsequent steps were performed as described above.

**Competition ELISA**. For bNAb competition ELISA experiments, rabbit sera (1:100 dilution) were incubated for 30 min with D7324-tagged ConM SOSIP.v7 that was captured onto solid phase using a D7324 tag. Human bNAbs were then added at a concentration to give sufficient signal (optical density 0.8–2.0) without competitor (0.1 μg/mL for PGT121, PGT128, and PGT135; 0.2 μg/mL for VRC01, VRC26.25, and PG16; 0.6 μg/mL for PG9 and PGT145; 1.0 μg/mL for 3BC315 and 35O22; 5.0 μg/mL for CH01 and gl-PG9; 10 μg/mL for gl-CH01). The bound bNAb was detected using horseradish peroxidase-labeled donkey anti-human IgG conjugate that was minimally cross-reactive with rabbit IgG (Jackson Immunoresearch, Westgrove, PA). The remaining binding signal of each well was subtracted from the negative control that contained serum but no bNAb (set to 0%) and compared to binding signal of the wells with bNAb only (set to 100%)[3].

**BLI and thermostability assay**. ConM SOSIP.v7 trimer or ConM SOSIP.v7-ferritin (1000 nM equimolar Env in PBS) were incubated for 1 h at 4 °C, 37 °C, 60 °C, 68 °C, or 72 °C. After 1 h, samples were diluted to 100 nM in BLI running buffer (PBS/0.1% bovine serum albumin/0.02% Tween20) and antibody binding was assessed using a ForteBio Octet K2. The binding assays were performed at 30 °C and with agitation set at 1000 rpm. Antibody was loaded on protein A sensors (ForteBio) at 2.0 μg/mL in running buffer until a binding threshold of 0.5 nm was reached. Association and dissociation were measured for 300 s. Binding to a protein A sensor not loaded by antibody was set as background.

**Negative-stain electron microscopy**. SOSIP trimers were imaged and processed as described previously[16]. In short, a 3-μL aliquot containing ~0.03 mg/mL of trimer or nanoparticles were applied for 5 s onto a carbon-coated 400 Cu mesh grid that had been glow discharged at 20 mA for 30 s, then negatively stained with 2% (w/v) uranyl formate for 60 s. Data were collected on a FEI Tecnai T12 electron microscope operating at 120 keV.

**Differential scanning calorimetry**. Thermal denaturation was studied using a nano-DSC calorimeter (TA instruments, Etten-Leur, The Netherlands)[16]. In short, ConM SOSIP.v7 trimer or ConM SOSIP.v7-ferritin were first dialyzed against PBS, and concentration was then adjusted to ~0.25 or ~1.0 mg/mL, respectively. After loading the sample into the cell, thermal denaturation was probed at a scan rate of 60 °C/h. Buffer correction, normalization, and baseline subtraction procedures were applied before the data were analyzed using the NanoAnalyze Software v.3.3.0 (TA Instruments). The data were fitted using a non-two-state model, as the asymmetry of some of the peaks suggested that unfolding intermediates were present. DSC experiments were performed with a D7324-tagged trimer, but the presence of the D7324-tag did not alter the $T_m$ values compared to the corresponding non-tagged trimers[16].

**Dynamic light scattering (DLS)**. DLS measurements were performed at 20 °C using a Dynapro Nanostar instrument (Wyatt Technologies), with 10 acquisitions of 5 s each. Each sample was centrifuged at 10,000 × g for 10 min prior to the DLS measurements to remove any trace of aggregates or dust from the sample. The hydrodynamic radii ($R_h$) were calculated using the Dynamics Analysis software (Wyatt Technologies), assuming a spherical model.

**Glycan profiling (hydrophilic interaction liquid chromatography-ultra performance liquid chromatography (HILIC-UPLC))**. Glycan profiling was done as previously described[18]. In short, SOSIP trimers were resolved by SDS-PAGE. Bands corresponding to gp140 were excised and the N-linked glycans attached to gp140 protein were released by N-glycosidase F (PNGase F; NEB). Glycans were labeled with 2-aminobenzoic acid. Fluorescently labeled glycans were resolved by HILIC-UPLC using a 2.1 × 10 mm² Acquity BEH Amide Column (1.7 μm particle size) (Waters, Elstree, UK). Fluorescence was measured using an excitation wavelength of 250 nm and a detection wavelength of 428 nm. Data processing was performed using the Empower 3 software (Waters Corporation, Milford, MA, USA). The percentage abundance of oligomannose-type glycans was calculated by integration of the relevant peak areas before and after Endoglycosidase H (EndoH) digestion, following normalization.

**Crystallization and data collection**. For complex formation, ConM SOSIP.v7 trimers, Fab PGT124, and Fab 35O222 were mixed in a 1:3.2:3.2 molar ratio and treated with EndoH (New England Biolabs) in 200 mM NaCl and 50 mM sodium citrate pH 5.5 for 45 min at 37 °C to decrease the heterogeneity of trimer:antibody complexes[31,32], followed by further SEC purification. SEC-purified ternary complexes were concentrated to ~6 mg/mL and screened against 480 crystallization conditions at 22 °C and 4 °C using our robotic CrystalMation system (Rigaku) at TSRI. Initial crystals of ConM SOSIP.v7 in complex with Fabs PGT124 and 35O22 were grown in JCSG Core II Suite (QIAGEN) condition G6 (0.07 M sodium acetate, 30% (v/v) glycerol, 5.6% (w/v) PEG 4000, pH 4.6), and after further optimization, crystals for data collection were obtained in 0.07 M sodium acetate, 30% (v/v) glycerol, and 6% (w/v) PEG 4000, pH 3.93. Crystals were flash cooled in liquid nitrogen since the crystallization condition already included a cryoprotectant. Diffraction data were collected at Advanced Photon Source beamline 23-ID-D and integrated/scaled with HKL-2000 to 3.9 Å in space group P6₃ with unit cell parameters $a = b = 127.6$ Å, $c = 315.5$ Å (Table 2).

**Structure determination and refinement**. The ternary complex structure was solved by molecular replacement method with Phaser[74] using the crystal structure of BG505 SOSIP.664 in complex with Fabs 3H+109L and 35O22 (PDB ID: 5CEZ) as a phasing model. After inserting the amino acid substitutions present in ConM SOSIP.v7 and PGT124, the initial structure was refined, using the LORESTR pipeline[75], and model building was performed using Coot[76] and refined with Refmac5[77]. The final $R_{work}$ and $R_{free}$ values were 24.7% and 29.6%, respectively, and residues in the Ramachandran favored and outliers region were 96.1% and 0.4%, respectively (Table 2). The Fab residues were numbered according to Kabat et al.[78] and gp120-gp41 residues using HxB2 numbering. Ramachandran statistics were calculated with MolProbity[79].

**B cell activation**. WEHI-231 B cell lines expressing the specific bNAbs (VRC01, PGT145, PGT121, and PG16) were donated by Takayuki Ota and David Nemazee at The Scripps Research Institute and were maintained in advanced Dulbecco's Modified Eagle's Medium (DMEM) supplemented with 10% FCS, β-mercapto-ethanol (55 μM), L-glutaMAX (2 mM), penicillin (100 U/mL), and streptomycin (100 μg/mL) (advanced DMEM++). The activation experiments and calcium flux measurements were performed as described elsewhere[26,47]. In short, 1 day prior to the assay, cells were incubated with 1 μg/mL doxycycline to induce the expression of BCRs. The next day, cells were suspended at 4 million cells/mL in advanced DMEM++, labeled with 1.5 μM Indo-1 (Invitrogen) for 30 min at 37 °C and washed with HBSS (Hank's Balance Salt Solution) containing 2 mM CaCl₂, followed by another incubation of 30 min at 37 °C. Aliquots of 1 million cells/mL were then stimulated at room temperature with SOSIP trimers, nanoparticles, or control reagents. Calcium (Ca²⁺) signals were recorded on a LSR Fortessa (BD Biosciences) by measuring for 210 s the 405/485 nm emission ratio of Indo-1 fluorescence upon UV excitation. ConM SOSIP.v7 and ConM SOSIP.v7-ferritin were tested at concentrations of 50 nM Env equivalent, i.e., 6.25 nM of ferritin nanoparticles. Kinetic analyses were performed using FlowJo v10.

**Immunizations**. In a first immunization study, 10 rabbits (New Zealand White, female, 2 groups, 5 animals/group) were immunized under subcontract at Covance (Denver, USA) with either 22 μg (Env protein content only; i.e., glycans are ignored in recording amounts) of PGT145-purified ConM SOSIP.v7 or ConM SOSIP.v7-ferritin formulated in 75 units of ISCOMATRIX™ adjuvant (CSL Ltd., Parkville, VIC, Australia) by two intramuscular immunizations in each quadriceps at weeks 0, 4, and 20. All immunization procedures complied with all relevant ethical regulations and protocols of the Covance Institutional Animal Care and Use Committee (IACUC). In a second study, 12 other rabbits (New Zealand White, female, 6 animals/group), at the Central National Food Chain Safety Office, Directorate of Veterinary Medicinal Products (NFCSO-DVMP, Gödöllő, Hungary), received either 20 μg of PGT145-purified ConM SOSIP.v7 or ConM SOSIP.v7-ferritin formulated inSE (Polymun, Klosterneuburg, Austria) by two intramuscular immunizations in each quadriceps at weeks 0, 4, and 20. Antigen was mixed 1:1 with SE (i.e., 250 μL of antigen in PBS combined with 250 μL SE). All procedures complied with all relevant ethical regulations for animal testing of the animal ethics committee of NFCSO-DVMP.

Rhesus macaques (*Macaca mulatta*, female, 1 group, 6 animals) received a PGT145-purified SOSIP-ferritin mix (trivalent cocktail with ConM SOSIP.v7-ferritin, AMC008 SOSIP.v5.2-ferritin and AMC011 SOSIP.v5.2-ferritin; 33 μg protein mass each) formulated in adjuvant (AddaVax, InvivoGen, San Diego, USA) by three intramuscular immunization (left arm, right arm, left thigh) at weeks 0, 4,

and 20. Two control macaques received adjuvant only. Macaque immunizations were carried out at the Biomedical Primate Research Centre (BPRC, Rijswijk, The Netherlands). All procedures and protocols complied with all relevant ethical regulations for animal testing of BPRC's Animal Experiments Committee.

**Neutralization assay**. TZM-bl and U87 neutralization assays were performed essentially as described elsewhere[80]. The assays were performed at four different sites: AMC, Academic Medical Center, Amsterdam, The Netherlands; DUMC, Duke University Medical Center, Durham, NC, USA; OSR, San Raffaele Scientific Institute, Milan, Italy; and ISCIII, Instituto de Salud Carlos III, Madrid, Spain. The TZM-bl reporter and U87 cell line were obtained through the NIH AIDS Research and Reference Reagents Program, Division of AIDS, NIAID, NIH (John C. Kappes, Xiaoyun Wu and Tranzyme Inc., Durham, NC, USA and HongKui Deng and Dan R. Littman, respectively). The $ID_{50}$ values were determined as the sera dilution at which infectivity was inhibited by 50%.

To determine the specificity of the NAb responses against ConM, sera were first incubated for 1 h at room temperature with 40 µg/mL of a depletion reagent. Subsequent steps of the neutralization assay were performed as described. ConM and BG505 have identical V3 sequences and the V3 peptide depletion reagent was as described before (TRPNNNTRKSIRIGPQAFYATGDIIGDIRQAH)[3]. The ConM SOSIP.v7 trimer with the swapped V1V2 was constructed by replacing the V1V2 of ConM (residues 131–196 in HXB2 numbering) with that of BG505[4]. All ConM SOSIP.v7 depletion reagents contain the D368R mutation to abrogate binding to CD4 on the TZM-bl receptor cell line and were positively selected by PGT145 affinity chromatography to ensure that only native-like trimers were used in the depletion experiments.

**HIV-1 Env amino acid conservation**. We used 6112 HIV-1 Env protein sequences from the Los Alamos HIV database (http://www.hiv.lanl.gov/) as present in 2017 to determine amino acid frequencies of HIV-1 Env residues. All HIV-1 Env protein sequences were aligned with the Jalview program and amino acid propensities per residue were calculated according to the alignment result from Jalview with 6112 Env protein sequences.

**Statistical analyses**. Groups were compared using unpaired two-tailed Mann–Whitney $U$ test unless noted otherwise. Spearman's rank correlation coefficient was used to determine correlations. All statistical analyses were performed in Graphpad Prism 7.0.

**Reporting summary**. Further information on research design is available in the Nature Research Reporting Summary linked to this article.

## Data availability

The coordinates and structure factors for the ConM SOSIP.v7 structure in complex with Fab PGT124 and Fab 35O22 have been deposited in the Protein Data Bank (PDB code 6IEQ). All other data supporting the findings in this manuscript are available from the corresponding authors (R.W.S., B.W.H., and I.A.W.) upon reasonable request.

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

## Acknowledgements

We thank Larry Liao and Bart Haynes for donating the DNA plasmid for generating ConS pseudovirus; Takayuki Ota and David Nemazee for kindly supplying the PG16, PGT145, VRC01, and PGT121 B cell lines; and Michel Nussenzweig, Hermann Katinger, Mark Connors, James Robinson, Dennis Burton, John Mascola, Peter Kwong, and William Olson for donating antibodies and reagents directly or through the NIH AIDS Research and Reference Reagent Program. We thank Hansi Dean, Wayne Koff, Joanne Stefano, and Beth Rasmussen for their contributions to rabbit study C0119–15. We thank Dietmar Katinger for providing the squalene emulsion adjuvant and Atilla Farsang and Réka Lévai for their contributions to rabbit study EAVI2020-II. We thank Sylvie Koekkoek for technical support. This project has received funding from the European Union's Horizon 2020 research and innovation program under grant agreement No. 681137 (to R.S., R.W.S., G.S., M.C., and J.A.). This work was also supported by the U.S. National Institutes of Health Grant P01 AI110657 (to J.P.M., A.B.W., I.A.W., and R.W. S.) and NIAID Contract #HHSN27201100016C (to D.C.M.); by the International AIDS Vaccine Initiative (IAVI); by the Bill and Melinda Gates Foundation through the Collaboration for AIDS Vaccine Discovery (CAVD), grants OPP1111923 and OPP1132237 (to J.P.M. and R.W.S.) and OPP1115782 (A.B.W.); by the Aids Fonds Netherlands, Grant #2016019 (to R.W.S.); and by the Fondation Dormeur, Vaduz (to R.W.S. and to M.J.v. G.). R.W.S. is a recipient of a Vici grant from the Netherlands Organization for Scientific Research (NWO). This work was partially supported by the Spanish Plan Nacional R+D +I [RD16/0017/0037] and FIS [PI16/1355], co-financed by ISCIII-Subdirección General de Evaluación y el Fondo Europeo de Desarrollo Regional (FEDER). This work was also supported by the Global Frontier Project (grant number: NRF-2013M-3A6A-4043695) and the Tumor Microenvironment Global Core Research Center (grant number: 2011–0030001) funded through the National Research Foundation from the Ministry of Science and ICT of Korea (to B.W.H). M.J.v.G. is a recipient of an AMC Fellowship and a Mathilde Krim Fellowship from the American Foundation for AIDS Research (amfAR) (109514–61-RKVA). J.MC-S. is a recipient of a fellowship from the Consejo Nacional de Ciencia y Tecnología of Mexico (CONACYT). The electron microscopy data were collected at Electron Microscopy Facility of The Scripps Research Institute. The Amsterdam Cohort Studies on HIV infection and AIDS, a collaboration between the Amsterdam Health Service, the Academic Medical Center of the University of Amsterdam, Sanquin Blood Supply Foundation, Medical Center Jan van Goyen and the HIV Focus Center of the DC-Clinics, are part of the Netherlands HIV Monitoring Foundation and financially supported by the Center for Infectious Disease Control of the Netherlands National Institute for Public Health and the Environment. X-ray data sets were collected at the Advanced Photon Source, Argonne National Laboratory (beamline 23 ID-D). GM/CA CAT is funded in whole or in part with federal funds from the National Cancer Institute (Y1-CO-1020) and NIGMS (Y1-GM-1104). Use of the Advanced Photon Source was supported by the U.S. Department of Energy (DOE), Basic Energy Sciences, Office of Science, under contract no. DE-AC02–06CH11357.

## Author contributions

Conceived and designed the experiments: K.S., B.W.H., I.B., F.G., R.J.S., I.A.W., R.W.S. Performed the experiments: K.S., B.W.H., I.B., F.G., Y.H., S.K., A.S., P.M., A.-J.B., K.R., E.S., M.T., J.L.T., G.O., P.v.d.W., A.T.d.l.P., M.J.v.B., J.M.C.-S., J.A.B., N.G. Provided

reagents: M.M.-R., P.B. Analyzed the data: K.S., B.W.H., I.B., F.G., S.K., A.S., A.-J.B., K.R., G.O., J.A.B., M.M.-R., N.G., J.A., C.L., G.S., M.J.v.G., M.C., D.C.M., A.B.W., G.K., I.A.W., R.W.S. Wrote the paper: K.S., I.B., B.W.H., I.A.W., R.W.S. Edited the paper: P.M., F.G., S.K., A.S., G.O., J.A., C.L., G.S., M.J.v.G., M.C., D.C.M., A.B.W., G.K., J.P.M., R.J.S., W.M.B.

## Additional information

**Competing interests:** The authors declare no competing interests.

