## [Peer Review File · Nature Communications]

Reviewers' Comments:

Reviewer #1:

Remarks to the Author:

Bontjer et al. describe in the current study an advanced strategy to generate stabilized HIV-1 trimers. The authors developed ConM, a stabilized HIV trimer based on a published group M consensus sequence. The authors generated first a stabilized trimer based on the SOSIP.v4.2 design. As this yielded unsatisfactory results the authors advanced the SOSIP design further leading up to the SOPSI.v7 design which incorporates additional changes that improve quality, stability and yield. Independent of the ConM Env that is the focus of the current study, the .v7 SOSIP design is interesting in itself, as it may provide a universal platform for generating well-folded soluble trimers. The presented work is the result of a large collaboration and, as previous work from these groups, is of high quality and provides detailed insight. The main short coming of the study is that despite all the refinement in protein design and structural insight, the ConM-SOSIP is not a convincing immunogen in terms of eliciting neutralizing responses. The achieved neutralizing activity is very modest.

Specific comments:

1. Lines 310-312: "Heterologous NAb responses against the Tier 1A clade B virus SF162 were weak in the trimer group, consistent with the presence of substitutions in the SOSIP.v7 design that limit exposure of non- neutralizing epitopes". Some questions to this data: The authors focus on ConM and the SF162 but do not discuss the response against the clade C virus MW965.26 (Supplementary Table 4a). This needs to be discussed as it was strongest activity next to the autologous ConM response. Considering that MW965.26 is a Tier 1A virus it would be very informative to verify what nAb specificity mediates the response. A nAb depletion experiment similar to Supplementary Table 5/ Fig. 5e and/or mutational analysis similar to Supplementary Table 6a,b/ Fig. 5f would provide helpful insights.
2. Lines 152-153: "The BG505 SOSIP.v5.2 trimer interacted well with gl-PG9, gl-PG16 and gl-CH01, consistent with previous observations using BG505SOSIP.664 (Fig. 1g; 35)". The statement for gl-PG16 reactivity needs to be revised. BG505 SOSIP.v5.2 interaction with gl-PG16 shown in Fig.1g appears poor, especially compared to the BG505 SOSIP GT1.1 and to ConM SOSIP.v7.
3. Lines 235-236: "The polydispersity [of ConM SOSIP.v7-ferritin] was relatively high (>15%), consistent with the presence of incomplete particles." Why do the authors refer to a level >15%? The DLS polydispersity value for ConM SOSIP.v7-ferritin in Table 1 is much higher than this (59.9%)?
4. Lines 238-240: "ConM SOSIP.v7-ferritin displayed an increased thermostability compared to ConM SOSIP.v7 trimers ($T_m = 72.0^\circ\text{C}$ versus $T_m=67.8$, respectively) (Fig. 3e, Table 1)". The DSC curve of ConM SOSIP.v7-ferritin shows peak broadening and asymmetry which may indicate presence of denaturation intermediates. In the methods section (Lines 626-627) the authors actually make a statement to this issue: "The data were fitted using a non-two-state model, as the asymmetry of some of the peaks suggested that unfolding intermediates were present." Does this problem apply to ConM SOSIP.v7-ferritin? A comparison of the antigenic stability of the immunogens after heat-treatment should be provided. If intermediates occur this would imply that the ferritin constructs have partially open trimers.
5. Related to the potential intermediates in the ferritin preparation: ConM SOSIP.v7 trimer seems not to induce V3 responses (no activity to SF162). However, ConM SOSIP.v7-ferritin immunized animals have activity against SF162. The authors need to verify whether this is due to V3 responses which may have been generated through partially opened trimers (intermediates.... See above)
6. The authors show that the main response to ConM SOSIP.v7 is directed to the apex. It would be helpful for the reader to see a sequence comparison of the V1V2 region of the viruses probed in the neutralization assay.
7. The authors should include data to confirm that ConS is a Tier-2 virus as the provided reference does not indicate this.

8. Lines 332-334: "The PGT121 and PG9 competition correlated well (Spearman $r = 0.74$, $p = 0.0178$) (Fig. 5b), suggesting that ConM Abs were targeting a region that affected both PGT121 and PG9 epitopes." Figure 5b is a bit random (one high correlation, one low). A full correlation matrix would be better. Please state the type of correlation analysis conducted.
9. The authors should comment whether or not anti-ferritin responses were detected.
10. Lines 400-403: "Six animals received three intramuscular vaccinations at weeks 0, 4 and 20 of 33 μg of each nanoparticle construct (i.e. 100 μg nanoparticles in total) formulated in Addavax adjuvant (Fig. 6a)." The authors seem to refer here to the full mass of the constructs not the mass of the Env protein only (i.e. without ferritin and glycans) as they did in the rabbit experiments? Is this correct? Why is the amount used for immunization of the macaques lower than for rabbits?
11. Figure 5d is misleading. The activity of the ConM immunogens against the ConM virus is shown twice (once in autologous, once in the ConM section) but BG505 neutralization activity of the ConM immunogens is not shown. The authors should skip the "autologous" section and instead show the BG505 titers. Autologous reactivity could be highlighted by open symbols instead.
12. The analysis of the macaque sera should be expanded: Longitudinal ConM neutralization titers should be shown as for rabbit sera. To allow better comparison with the rabbit experiments, neutralization activity to TV1.21, AMC011, AMC008 and ConS mutants should be shown. V3 ELISA and neutralization competition experiments similar to what is shown for the rabbit sera would help to clarify if the response targets similar epitopes.

Reviewer #2:

Remarks to the Author:

Bontjer et al. have reported a SOSIP Env trimer, named ConM trimer, derived from a consensus sequence of all HIV-1 group M isolates (ConM). This trimer showed binding to the most known bNABs, consistent with the antigenicity of a native Env, as well as to the inferred germline precursors of several bNABs, potentially capable of activating relevant B cells. The authors have also determined the crystal structure of the trimer, which turned out to be very similar to those of other SOSIP Env trimers. Moreover, this trimer induced strong autologous NAb responses in both rabbits and macaques, in particular, when presented on ferritin nanoparticles. NABs in the sera of the immunized animals appeared to target a region close to the trimer apex. They concluded that immunogens based on consensus sequences might help vaccine development against HIV-1 and other viruses.

The manuscript is generally well-written, and the data are technically sound and well-presented. But it reads more like a collection of some well-performed experiments with only incremental advances over what the authors have published previously. Since the SOSIP design was first published in 2013, there have been high hopes in the field that the SOSIP trimers may induce potent bNAB responses in animal models and eventually in humans, but so far they only induced some autologous responses that are difficult to broaden (even if it is a tier 2 response). Nevertheless, several groups have been working on strategies for further stabilizing and better producing these trimers (e.g. several versions of SOSIP mentioned in the manuscript) and have already published many papers with no significant improvement in the immunogenicity. Using consensus sequences to make HIV-1 Env-based vaccines is not new. It was certainly worth trying them on the SOSIP design, but the immunogenicity data presented here remain disappointing despite all perceived advantages (native-like antigenicity and ability to engage germline antibodies) and they do not seem to have taught us much about future directions for vaccine design. There is no question that these data should be published somewhere as a nice record that these experiments have been done. Even for that purpose, it would be nice to compare the current trimer with the previous consensus sequence-based Env immunogens.

The authors stated that "Moreover, the majority of the glycans on ConM SOSIP.v7 trimers are

oligomannose (68%), in particular Man₉GlcNAc₂ (25%) and Man₈GlcNAc₂ (12%) (Fig. 1f, Table 1), which is similar to viral Env and other soluble native-like Env trimers". A recent report (Cao et al., Nat Commun. 2018 Sep 12;9(1):3693) showed that "site-specific glycosylation of Env from infectious virus closely matches Envs from corresponding recombinant membrane-bound trimers. However, viral Envs differ significantly from recombinant soluble, cleaved (SOSIP) Env trimers, strongly impacting antigenicity". The authors might want to comment on the different claims.

It's hard to follow the logic for claiming that the ConM trimer displays few 'rare' residues. Consensus sequences, by definition, are not rare and the ConM trimer would have fewer rare residues found in all other natural isolates. The authors mentioned that "ConM residues with less than 30% amino acid conservation among the sequences from the Los Alamos database were considered as 'rare' residues. Second, 'rare' residues of ConM with less amino acid conservation than equipositional 'rare' residues of BG505 SOSIP (as defined by a difference >3%) were considered as ConM-specific rare residues and vice versa for BG505 SOSIP-specific rare residues". It is unclear why 30% and 3% were chosen to define "rareness". In fact, these numbers could be totally arbitrary but they would have a big impact on the number of residues that fit the criteria. If that's that case, this so-called amino acid conservation analysis would probably not be very meaningful.

Response to reviewers

Reviewer #1

Bontjer et al. describe in the current study an advanced strategy to generate stabilized HIV-1 trimers. The authors developed ConM, a stabilized HIV trimer based on a published group M consensus sequence. The authors generated first a stabilized trimer based on the SOSIP.v4.2 design. As this yielded unsatisfactory results the authors advanced the SOSIP design further leading up to the SOPSI.v7 design which incorporates additional changes that improve quality, stability and yield. Independent of the ConM Env that is the focus of the current study, the .v7 SOSIP design is interesting in itself, as it may provide a universal platform for generating well-folded soluble trimers.

Many thanks for the supportive comments. We indeed believe that the new SOSIP.v7 design provides a new and universal platform for generating soluble trimers. We had not emphasized this aspect in the previous version of the paper, but we have now added a sentence to the discussion section to do so (lines 493-494).

The presented work is the result of a large collaboration and, as previous work from these groups, is of high quality and provides detailed insight. The main short coming of the study is that despite all the refinement in protein design and structural insight, the ConM-SOSIP is not a convincing immunogen in terms of eliciting neutralizing responses. The achieved neutralizing activity is very modest.

The neutralizing antibody responses that the ConM SOSIP trimer induces are actually quite strong (titers in the 10,000-100,000 range for the autologous virus and 100-1,000 range for the related ConS virus). Perhaps the comment is then related to the breadth in specificity, which we admit is not broad. However, this is not unexpected as the induction of broadly neutralizing activity during natural infection is the product of a coevolutionary process that requires multiple antigenic variants to collectively lead to the development of neutralization breadth. We therefore did not *a priori* expect ConM SOSIP to induce neutralization breadth when used as a stand-alone immunogen, but rather propose it to be a component of a vaccine that induces broadly neutralization antibodies. For example, it could be the “polishing” immunogen that finalizes the antibody maturation process of antibody lineages that were primed by a germline-targeting vaccine (see for example references 41 and 44). We have now made this point more clear at the end of the discussion section (lines 549-557).

Specific comments:

1. Lines 310-312: “Heterologous NAb responses against the Tier 1A clade B virus SF162 were weak in the trimer group, consistent with the presence of substitutions in the SOSIP.v7 design that limit exposure of non- neutralizing epitopes”. Some questions to this data: The authors focus on ConM and the SF162 but do discuss the response against the clade C virus MW965.26 (Supplementary Table 4a). This needs to be discussed as it was strongest activity next to the autologous ConM response. Considering that MW965.26 is a Tier 1A viruses it would be very informative to verify what nAb specificity mediates the response. A nAb depletion experiment similar to Supplementary Table 5/ Fig. 5e and/or mutational analysis similar to Supplementary Table 6a,b/ Fig. 5f would provide helpful insights.

Indeed, the neutralization activity against MW965.26 was very strong. We have now performed V3 peptide depletion experiments with MW965.26, SF162, as well as with Tier 2 virus TV1.21. The results show that the MW965.26 and SF162 neutralization (in contrast to ConM neutralization) is partly mediated by V3-specificities, in particular in the ferritin nanoparticle group, an observation that is probably related to this reviewers’ points 4 and 5 (see below). The new data can be found in Supplementary Table 7. We have also modified the accompanying results section to incorporate these new data (lines 342-350).

2. Lines 152-153: “The BG505 SOSIP.v5.2 trimer interacted well with gl-PG9, gl-PG16 and gl-CH01, consistent with previous observations using BG505SOSIP.664 (Fig. 1g; 35)”. The statement for gl-

PG16 reactivity needs to be revised. BG505 SOSIP.v5.2 interaction with gl-PG16 shown in Fig.1g appears poor, especially compared to the BG505 SOSIP GT1.1 and to ConM SOSIP.v7.

We agree and have modified the text accordingly (lines 151-152).

3. Lines 235-236: “The polydispersity [of ConM SOSIP.v7-ferritin] was relatively high (>15%), consistent with the presence of incomplete particles.” Why do the authors refer to a level >15%? The DLS polydispersity value for ConM SOSIP.v7-ferritin in Table 1 is much higher than this (59.9%)?

In Dynamic Light Scattering (DLS) experiments, 15% is the standard cut-off for monodispersity. A value of <15% is considering monodisperse and indicative of highly homogeneous particles. A value of >15% is indicative of heterogeneity. The definition of this cut-off originates from Habel *et al.* 2001, *Acta Cryst.* This reference is now included (reference 53).

4. Lines 238-240: “ConM SOSIP.v7-ferritin displayed an increased thermostability compared to ConM SOSIP.v7 trimers ($T_m = 72.0^\circ\text{C}$ versus $T_m=67.8$, respectively) (Fig. 3e, Table 1)”. The DSC curve of ConM SOSIP.v7-ferritin shows peak broadening and asymmetry which may indicate presence of denaturation intermediates. In the methods section (Lines 626-627) the authors actually make a statement to this issue: “The data were fitted using a non-two-state model, as the asymmetry of some of the peaks suggested that unfolding intermediates were present.” Does this problem apply to ConM SOSIP.v7-ferritin? A comparison of the antigenic stability of the immunogens after heat-treatment should be provided. If intermediates occur this would imply that the ferritin constructs has partially open trimers.

We thank the reviewer for this insightful comment. Indeed, the ConM SOSIP-ferritin construct has a broader and more asymmetric peak. However, unfolding does not start earlier than for ConM SOSIP trimers. We now see that this was not obvious from the presentation of our data and have now generated a plot with an overlay of the unfolding curves for both ConM trimers and ferritin particles (Fig. 3e). We indeed think that a small proportion of the trimers on the ConM SOSIP-ferritin are non-native and partially open (see also response to point 5), probably corresponding to the uncleaved fraction (now labeled specifically as such in Supplementary Figure 5b).

We appreciate the suggestion of performing heat-treatment experiments to further probe the properties of the free trimers and nanoparticle-associated trimers. We therefore performed entirely new sets of experiments using Biolayer Interferometry (BLI). The antigenicity results from those studies are valuable in themselves (new Supplementary Fig. 6) as they confirm and extend the antigenicity data that we included in the previous version and that were obtained by ELISA (Supplementary Fig. 5a), but we also used them to perform the requested heat-treatment experiments. Thus, we incubated both trimers and nanoparticles for an hour at 4°C (negative control), 37°C (mimicking vaccination), and at 60°C, 68°C and 72°C (mirroring the unfolding events in the DSC profile), prior to antigenicity analysis using BLI. At 4°C and 37°C bNAbs bound more strongly to the ferritin nanoparticles, but non-NABs did so as well (see next point). For both trimers and nanoparticles bNAb binding decreased dramatically after incubation at 68°C or 72°C, consistent with the thermal unfolding observed in the DSC. In contrast, binding of the V3 non-NAb 19b increased after incubation at higher temperatures, consistent with its linear epitope becoming exposed during unfolding. F105 binding first increased (60°C) then decreased upon increasing the temperature (68°C and 72°C), consistent with a conformational epitope that becomes exposed when the trimer dissociates, but is then damaged when the individual protomers unfold. These data are presented in the new Fig. 3f and the raw data can be found in Supplementary Fig. 6. The data are described in the results (lines 240-270). A methods section for BLI was also included (lines 679-687). We moved the raw NS-EM images in the original Fig. 3b to Supplementary Fig. 5c to make space for Fig. 3f.

5. Related to the potential intermediates in the ferritin preparation: ConM SOSIP.v7 trimer seems not to induce V3 responses (no activity to SF162). However, ConM SOSIP.v7-ferritin immunized animals have activity against SF162. The authors need to verify whether this is due to V3 responses which may have been generated through partially opened trimers (intermediates.... See above)

We have verified whether these responses and also those directed against MW965.26 and two TV1.21 mutants were V3-mediated (see also response to this reviewer's point 1), and they indeed were, at least partially (see Supplementary Table 7). The exposure of V3 was also confirmed by BLI (new Fig. 3 and Supplementary Fig. 6). We postulate that the incomplete cleavage of the trimers on ferritin results in the presence of a minority of uncleaved, nonnative, and partially open trimers that cause the induction of V3-specificities. This is explained in the text (lines 342-350).

6. *The authors show that the main response to ConM SOSIP.v7 is directed to the apex. It would be helpful for the reader to see a sequence comparison of the VIV2 region of the viruses probed in the neutralization assay.*

Thanks- we have now included such a comparison and inserted it in panel Fig. 5f.

7. *The authors should include data to confirm that ConS is a Tier-2 virus as the provided reference does not indicate this.*

The ConS virus has been used in several papers and designated to be a Tier 2 virus (Liao *et al.* 2006, *Virology*; Andersson *et al.* 2016, *Vaccine*; Aldon *et al.* 2018, *Cell Rep.*). The designation was based on historical Tier categorization data from the neutralization reference lab of David Montefiori, an author on this paper. In response to the reviewer's comment, the ConS virus was again analyzed using newer Tier categorization reagents (see new Supplementary Table 4) and reclassified as Tier 1B, albeit at the resistant end of the Tier 1B spectrum. We have edited the text and labels accordingly.

8. *Lines 332-334: "The PGT121 and PG9 competition correlated well (Spearman $r=0.74$, $p=0.0178$) (Fig. 5b), suggesting that ConM Abs were targeting a region that affected both PGT121 and PG9 epitopes." Figure 5b is a bit random (one high correlation, one low). A full correlation matrix would be better. Please state the type of correlation analysis conducted.*

We thank the reviewer for this useful suggestion and have now included a Spearman correlation matrix to support the competition ELISA results (new Supplementary Fig. 8). The results show that binding to bNAb PGT128 correlated even better with VIV2 bNAb binding than did binding to PGT121. We have adjusted Fig. 5b (two examples of PGT128 correlation) and Fig. 5f to reflect this.

9. *The authors should comment whether or not anti-ferritin responses were detected.*

We have now tested rabbit and macaque sera for anti-ferritin and found that such responses are induced, as expected. This is in agreement with previous reports on influenza HA ferritin nanoparticles (reference 61). The results are shown in the new Supplementary Fig. 9 and we have made several additions to the methods section (lines 593-595, lines 641-644, lines 664-667) and refer to these data in the results (lines 448-453) and discussion (lines 526-528).

10. *Lines 400-403: "Six animals received three intramuscular vaccinations at weeks 0, 4 and 20 of 33 μg of each nanoparticle construct (i.e. 100 μg nanoparticles in total) formulated in Addavax adjuvant (Fig. 6a)." The authors seem to refer here to the full mass of the constructs not the mass of the Env protein only (i.e. without ferritin and glycans) as they did in the rabbit experiments? Is this correct? Why is the amount used for immunization of the macaques lower than for rabbits?*

Thanks- we now realize that our original phrasing was ambiguous. The macaques were immunized at each timepoint with 33 μg of each construct in a trivalent cocktail (i.e. a 100 μg protein dose (ferritin + Env) at each immunization). The values indeed do not reflect the full mass as the glycans are excluded. We have now clarified the text in the manuscript (lines 438 and 788).

11. *Figure 5d is misleading. The activity of the ConM immunogens against the ConM virus is shown twice (once in autologous, once in the ConM section) but BG505 neutralization activity of the ConM immunogens is not shown. The authors should skip the "autologous" section and instead show the BG505 titers. Autologous reactivity could be highlighted by open symbols instead.*

We have modified Fig. 5d and its legend accordingly.

12. *The analysis of the macaque sera should be expanded: Longitudinal ConM neutralization titers should be shown as for rabbit sera. To allow better comparison with the rabbit experiments, neutralization activity to TV1.21, AMC011, AMC008 and ConS mutants should be shown. V3 ELISA and neutralization competition experiments similar to what is shown for the rabbit sera would help to clarify if the response targets similar epitopes.*

We have extended the analysis of the macaque sera with the suggested additional experiments. First, we performed ConM neutralization assays on longitudinal samples (new Fig. 6c). Furthermore, we performed neutralization depletion experiments to assess the specificity of the ConM NAb response (new Fig. 6d), showing that the V1V2 domain is targeted in both rabbits and macaques. In addition, we mapped the macaque ConS NAb epitopes using the panel of ConS mutants (new Fig. 6f and Supplementary Table 9) and performed neutralization assays using a panel of 20 Tier 1 and Tier 2 viruses, including TV1.21, AMC011 and AMC008 (Supplementary Table 8). Overall, these data confirm and strengthen the earlier observations made using the rabbit sera. The new results are described in lines 450-468.

Reviewer #2

Bontjer et al. have reported a SOSIP Env trimer, named ConM trimer, derived from a consensus sequence of all HIV-1 group M isolates (ConM). This trimer showed binding to the most known bNAbs, consistent with the antigenicity of a native Env, as well as to the inferred germline precursors of several bNAbs, potentially capable of activating relevant B cells. The authors have also determined the crystal structure of the trimer, which turned out to be very similar to those of other SOSIP Env trimers. Moreover, this trimer induced strong autologous NAb responses in both rabbits and macaques, in particular, when presented on ferritin nanoparticles. NAbs in the sera of the immunized animals appeared to target a region close to the trimer apex. They concluded that immunogens based on consensus sequences might help vaccine development against HIV-1 and other viruses.

We thank the reviewer for the succinct summary of our manuscript.

The manuscript is generally well-written, and the data are technically sound and well-presented. But it reads more like a collection of some well-performed experiments with only incremental advances over what the authors have published previously. Since the SOSIP design was first published in 2013, there have been high hopes in the field that the SOSIP trimers may induce potent bNAb responses in animal models and eventually in humans, but so far they only induced some autologous responses that are difficult to broaden (even if it is a tier 2 response). Nevertheless, several groups have been working on strategies for further stabilizing and better producing these trimers (e.g. several versions of SOSIP mentioned in the manuscript) and have already published many papers with no significant improvement in the immunogenicity.

SOSIP trimers were indeed the first immunogens to induce NAb responses against hard-to-neutralize viruses (Tier 2 viruses). This was a major breakthrough for the field. Sporadically, SOSIP trimers also induce broader responses but these sporadic broad responses tend to be weak. However, since the induction of bNAbs during natural infection is the product of a coevolutionary process that requires multiple antigenic variants that collectively lead to the development of neutralization breadth, it was probably naïve for the field to think that a single antigen - any single antigen - would induce bNAbs. We therefore did not *a priori* expect ConM SOSIP to induce neutralization breadth when used as a stand-alone immunogen, but rather propose it to be a component of a vaccine that induces broadly neutralization antibodies. For example, it could be the “polishing” immunogen that finalizes the antibody maturation process of antibody lineages were primed by a germline-targeting vaccine (see for example references 38 and 41). In such a setting, the ConM trimer might have an advantage over trimers derived from individual HIV-1 strains because it has much fewer strain-specific antigenic residues/determinants that are likely to distract from broad responses. We have now made this much more clear in various places in the text (lines 197-206 and lines 549-557). We also note that the ConM

SOSIP.v7 will be evaluated in two First-in-Human phase I trials in 2019/2020, which is testimony to the trust that funding agencies and partners have in the value of this immunogen.

Using consensus sequences to make HIV-1 Env-based vaccines is not new. It was certainly worth trying them on the SOSIP design, but the immunogenicity data presented here remain disappointing despite all perceived advantages (native-like antigenicity and ability to engage germline antibodies) and they do not seem to have taught us much about future directions for vaccine design.

We do not perceive the results described in this manuscript as disappointing. The NAb titers induced by the ConM SOSIP trimer are in the 10,000-100,000 range for the ConM virus and 100-1,000 range for the ConS virus. These are highly respectable titers and at the top-end of what has been observed across the field. We think that the strong immunogenicity of the ConM trimer bodes well for its use in lineage vaccines and germline approaches.

There is no question that these data should be published somewhere as a nice record that these experiments have been done. Even for that purpose, it would be nice to compare the current trimer with the previous consensus sequence-based Env immunogens.

We have performed extensive in vitro and in vivo comparisons of gp120, non-native gp140 trimers and SOSIP trimers (Sanders *et al.* 2015, *Science*; Ringe *et al.* 2013, *PNAS*; Ringe *et al.* 2015, *J.Virol.* Torrents de la Pena *et al.* 2017, *Cell Rep.*). The results are so clear cut and in favor of the SOSIP design that we felt it was unnecessary to do repeat such studies. The SOSIP design mimics the native Env trimer, whereas the earlier consensus Env immunogens did not. However, we have referred to earlier studies using consensus immunogens (introduction lines 85-88). We also extended the reference to our studies that compare SOSIP trimers with previous Env immunogen designs, i.e. gp120 and uncleaved gp140 trimers (introduction lines 62-65).

*The authors stated that “Moreover, the majority of the glycans on ConM SOSIP.v7 trimers are oligomannose (68%), in particular Man9GlcNAc2 (25%) and Man8GlcNAc2 (12%) (Fig. 1f, Table 1), which is similar to viral Env and other soluble native-like Env trimers”. A recent report (Cao *et al.*, *Nat Commun.* 2018 Sep 12;9(1):3693) showed that “site-specific glycosylation of Env from infectious virus closely matches Envs from corresponding recombinant membrane-bound trimers. However, viral Envs differ significantly from recombinant soluble, cleaved (SOSIP) Env trimers, strongly impacting antigenicity”. The authors might want to comment on the different claims.*

Although the oligomannose glycans are marginally higher than observed in membrane-tethered material (Cao *et al.* 2018 *Nat. Commun.*), they are consistent with a well-folded, native-like soluble trimer (Pritchard *et al.* 2015 *Cell Rep*; Behrens *et al.* 2016 *Cell Rep*; Sarkar *et al.* 2018 *Nat. Commun.*). We have now adjusted the text and added the Cao *et al.* 2018 *Nat. Commun.* reference.

It's hard to follow the logic for claiming that the ConM trimer displays few 'rare' residues. Consensus sequences, by definition, are not rare and the ConM trimer would have fewer rare residues found in all other natural isolates.

Indeed, and this summarizes the rationale for our study, as our aim was to decrease the number of rare amino acids on SOSIP trimer immunogens, and we thought this was worth stating for those that don't think much about consensus sequences.

The authors mentioned that “ConM residues with less than 30% amino acid conservation among the sequences from the Los Alamos database were considered as 'rare' residues. Second, 'rare' residues of ConM with less amino acid conservation than equipositional 'rare' residues of BG505 SOSIP (as defined by a difference >3%) were considered as ConM-specific rare residues and vice versa for BG505 SOSIP-specific rare residues”. It is unclear why 30% and 3% were chosen to define “rareness”. In fact, these numbers could be totally arbitrary but they would have a big impact on the number of residues that fit the criteria. If that's that case, this so-called amino acid conservation analysis would probably not be very meaningful.

We appreciate the reviewer's comment about our amino acid conservation analysis results. To further validate our amino acid conservation analysis results, we revisited and carefully re-analyzed our amino acid conservation data. There are 65 residues that exhibit different amino acid conservation rates between ConM and BG505 (new Supplementary Table 3). Among these 65 residues, ConM contains 14 rarer residues (see table below), while BG505 contains 51. As the reviewer worried about our “arbitrary” criteria for ‘rare’ residues, we applied various alternative criteria for ‘rare’ amino acids and the results are summarized in the table below. Overall, using all criteria, ConM contains considerably fewer ConM-specific rare residues than BG505. Thus, it is reasonable to conclude that the ConM SOSIP.v7 trimer displays fewer rare antigenic determinants compared to BG505, as it stands in our submitted manuscript. However, to minimize the impression that any given criteria for defining rare amino acids would yield arbitrary results, we now provide the residue numbers of amino acids in ConM or BG505 that rarer in one of them compared to the other. We revised our manuscript (lines 197-202) and Fig. 2c accordingly and added the Supplementary Table 3 to show the amino acid conservation analysis results.

Definition of rare amino acids , as percent amino acid conservation in natural HIV-1 isolates (Los Alamos database)	Definition of rarer amino acids when comparing the amino acid conservation of rare residues in ConM and BG505, as percent difference in conservation	Number of ConM-specific, rare residues	Number of BG505-specific, rare residues
<30%	>10%	5	44
<30%	>5%	6	46
<30%	>3%	6	48
<30%	>1%	9	48
<20%	>10%	3	39
<20%	>5%	4	40
<20%	>3%	4	42
<20%	>1%	7	42
<10%	>10%	2	27
<10%	>5%	2	27
<10%	>3%	2	29
<10%	>1%	5	29

Finally, we also performed new analyses (presented in the new Fig. 6d), in which we analyzed the presence of strain-specific glycan holes using a new tool (the Glycan Shield Mapping tool; see reference 49) that became available during review of this manuscript, and found that ConM, in contrast to BG505, has no strain-specific glycan holes (lines 202-205).

Reviewers' Comments:

Reviewer #1:

Remarks to the Author:

The authors have addressed all my queries.

Reviewer #2:

Remarks to the Author:

The authors have addressed some of my previous concerns in the revised manuscript. Certain issues remain, however. In particular, they still insisted that "the majority of the glycans on ConM SOSIP.v7 trimers are oligomannose (68%), in particular Man9GlcNAc2 (25%) and Man8GlcNAc2 (12%) (Fig. 1f, Table 1), which is similar to viral Env but exhibits marginally lower oligomannose levels, and very closely matches other soluble native like Env trimers³⁰⁻³⁶", despite the recent consensus in the field that "viral Envs differ significantly from recombinant soluble, cleaved (SOSIP) Env trimers" (Struwe et al., *Cell Rep.* 2018;24(8):1958-1966.e5; Behrens et al., *J Virol.* 2017;91(2):1-16; Cao et al, *Nat Commun.* 2018;9(1):3693). Indeed, the revised statement is even more confusing. Does it mean that the viral Envs have even higher oligomannose levels than 68%?

In addition, yes, the authors have performed extensive *in vitro* and *in vivo* comparisons of gp120, non-native gp140 trimers and SOSIP trimers, but those experiments were not performed using any consensus Env sequences. They are right that "The NAb titers induced by the ConM SOSIP trimer are in the 10,000-100,000 range for the ConM virus and 100-1,000 range for the ConS virus. These are highly respectable titers and at the top-end of what has been observed across the field". The problem is that the NAb titers by the ConM SOSIP trimer are ~1,000 against the virus SF162 (Fig. 4e; of course rabbits were used in these studies). Liao et al, has already reported NAb titers of 10,000-100,000 in guinea pigs when immunized with the group M consensus Env CON-S in a design called gp140CFI, which contains deletion of the cleavage (C) site and fusion (F) domain and deletions in the immunodominant (I) region in gp41, and is clearly non-native by the SOSIP standards (Table 5 in Liao et al., *Virology.* 2006 Sep 30; 353(2): 268-282; Fig 2 in Liao et al., *Journal of Virology* Mar 2013, 87 (8) 4185-4201). Without a side-by-side comparison, how can one be convinced that the ConM SOSIP trimer has improved immunogenicity over so-called nonnative Env immunogens?

Response to reviewers

(original reviewer's comments in *italics*, responses in blue)

Reviewer #1

The authors have addressed all my queries.

We are pleased that we accommodated the reviewer's concerns.

Reviewer #2

The authors have addressed some of my previous concerns in the revised manuscript.

We are pleased that we were able to address most of the reviewer's concerns.

Certain issues remain, however. In particular, they still insisted that “the majority of the glycans on ConM SOSIP.v7 trimers are oligomannose (68%), in particular Man9GlcNAc2 (25%) and Man8GlcNAc2 (12%) (Fig. 1f, Table 1), which is similar to viral Env but exhibits marginally lower oligomannose levels, and very closely matches other soluble native like Env trimers^{30–36}”, despite the recent consensus in the field that “viral Envs differ significantly from recombinant soluble, cleaved (SOSIP) Env trimers” (Struwe et al., Cell Rep. 2018;24(8):1958-1966.e5; Behrens et al., J Virol. 2017;91(2):1-16; Cao et al, Nat Commun. 2018;9(1):3693). Indeed, the revised statement is even more confusing. Does it mean that the viral Envs have even higher oligomannose levels than 68%?

To prevent any confusion arising, we have now deleted “is similar to levels”. The sentence now reads as follows: “the majority of the glycans on ConM SOSIP.v7 trimers are oligomannose (68%), in particular Man9GlcNAc2 (25%) and Man8GlcNAc2 (12%) (Fig. 1f, Table 1), which very closely matches other soluble native like Env trimers^{30–36}”

In addition, yes, the authors have performed extensive in vitro and in vivo comparisons of gp120, non-native gp140 trimers and SOSIP trimers, but those experiments were not performed using any consensus Env sequences. They are right that “The NAb titers induced by the ConM SOSIP trimer are in the 10,000-100,000 range for the ConM virus and 100-1,000 range for the ConS virus. These are highly respectable titers and at the top-end of what has been observed across the field”. The problem is that the NAb titers by the ConM SOSIP trimer are ~1,000 against the virus SF162 (Fig. 4e; of course rabbits were used in these studies). Liao et al, has already reported NAb titers of 10,000-100,000 in guinea pigs when immunized with the group M consensus Env CON-S in a design called gp140CFI, which contains deletion of the cleavage (C) site and fusion (F) domain and deletions in the immunodominant (I) region in gp41, and is clearly non-native by the SOSIP standards (Table 5 in Liao et al., Virology. 2006 Sep 30; 353(2): 268–282; Fig 2 in Liao et al., Journal of Virology Mar 2013, 87 (8) 4185-4201). Without a side-by side comparison, how can one be convinced that the ConM SOSIP trimer has improved immunogenicity over so-called nonnative Env immunogens?

We have now added the phrase “Although, we did not perform a direct comparison with ConM gp120 or nonnative ConM gp140 forms, ” to the third sentence of the discussion. The full sentence now reads: “Although, we did not perform a direct comparison with ConM gp120 or nonnative ConM gp140 forms, the ConM SOSIP.v7 trimer induced very strong autologous NAb responses against the autologous Tier 1A ConM virus in animals, especially when presented on ferritin nanoparticles, and induced substantial NAb responses against the related ConS virus.”